# Mechanically tunable organogels from highly charged polyoxometalate clusters loaded with fluorescent dyes

Fenghua Zhang [1], Zhong Li [1] ✉ & Xun Wang [1] ✉

Inorganic nanowires-based organogel, a class of emerging organogel with convenient preparation, recyclability, and excellent mechanical properties, is in its infancy. Solidifying and functionalizing nanowires-based organogels by designing the gelator structure remains challenging. Here, we fabricate $Ca_2$-$P_2W_{16}$ and $Ca_2$-$P_2W_{15}M$ nanowires utilizing highly charged $[Ca_2P_2W_{16}O_{60}]^{10-}$ and $[Ca_2P_2W_{15}MO_{60}]^{14-/13-}$ cluster units, respectively, which are then employed for preparing organogels. The mechanical performance and stability of prepared organogels are improved due to the enhanced interactions between nanowires and locked organic molecules. Compressive stress and tensile stress of $Ca_2$-$P_2W_{16}$ nanowires-based organogel reach 34.5 and 29.0 kPa, respectively. The critical gel concentration of $Ca_2$-$P_2W_{16}$ nanowires is as low as 0.28%. Single-molecule force spectroscopy confirms that the connections between cluster units and linkers can regulate the flexibility of nanowires. Furthermore, the incorporation of fluorophores into the organogels adds fluorescence properties. This work reveals the relationships between the microstructures of inorganic gelators and the properties of organogels, guiding the synthesis of high-performance and functional organogels.

Organogels exhibit numerous exceptional properties and multi-functionality, which are usually formed by locking organic molecules within three-dimensional (3D) networks constructed by gelators, such as microfibers, strands, or tapes through specific interactions (van der Waals forces, electrostatic forces, and hydrogen bonds, etc.)[1,2]. Reported organogels are prepared mainly by gelators of polymers and low-molecular-mass organogelators (LMOGs)[3–5]. Although the mechanical properties of these organogels are satisfactory, they are still hindered by complex preparation processes or challenging recovery methods. Recently, a class of gelators (inorganic nanowires (NWs) with a thickness of around 1 nm and polymer-like properties) has emerged, which exhibits compatibility with various organic liquids[6–8]. Although many NWs have been employed for gel preparation, their gelation performance is inadequate, often resulting in non-freestanding gels with poor mechanical properties.

So far, only one NW can be used to prepare freestanding, elastic, and stable organogels, alkaline earth metal cations-bridged polyoxometalate (POM) clusters nanowires (AE-$PW_{12}$ NWs)[9,10]. POM represents a class of atomically precise clusters featuring subnano or nano-size, adjustable charge, and high stability[11–13]. AE-$PW_{12}$ NWs are assembled by the electrostatic interactions of alkaline earth metal cations and $[PW_{12}O_{40}]^{3-}$ POM clusters, which can easily form 3D interwoven structures through electrostatic and van der Waals interactions, efficiently trapping organic solvents for easy recycling. However, the limited charge of the $PW_{12}$ cluster unit restricts the mechanical performance and functions of NW-based organogels constructed using it. In the NWs-based organogels, the interaction between the solvent molecules and surface ligands directly affects the performance of organogels[14,15]. Using the highly charged POM clusters as building blocks for constructing the NWs can effectively regulate their surface ligand density and then improve the mechanical

[1]Engineering Research Center of Advanced Rare Earth Materials, Department of Chemistry, Tsinghua University, Beijing, China.
✉e-mail: zhongli@mail.tsinghua.edu.cn; wangxun@mail.tsinghua.edu.cn

performance of NW-based organogels and get low critical gelation concentrations (CGC)[16]. Unfortunately, at present, the PW$_{12}$ cluster remains the only viable option for preparing the freestanding NW-based organogel.

Here, we synthesized a series of NWs assembled by the highly charged Dawson-type POM clusters ([Ca$_2$P$_2$W$_{16}$O$_{60}$]$^{10-}$ and [Ca$_2$P$_2$W$_{15}$MO$_{60}$]$^{14-/13-}$), called Ca$_2$-P$_2$W$_{16}$ NWs and Ca$_2$-P$_2$W$_{15}$M NWs (M = Fe, Mn, Ni, Co, Cr, Pr, Nd, Gd, Dy, Lu). Characterizations showed that these NWs have a thickness of one nanometer, good flexibility, and high surface ligand density. The organogels prepared from these NWs exhibit excellent mechanical performance and remain stable after two months of storage in a sealed container due to the enhanced interaction between NWs and solvent molecules. Especially, the compressive and tensile stress of the Ca$_2$-P$_2$W$_{16}$ NW-based organogel reached up to 34.5 and 29.0 kPa in the case of maximum strain, respectively. The CGC of Ca$_2$-P$_2$W$_{16}$ NW is only 0.28%. Single-molecule force spectroscopy (SMFS) reveals that the Ca$_2$-P$_2$W$_{16}$ NW has a 1.13 ± 0.31 nm persistence length (PL) due to the partly limited rotation caused by the connection of Ca$^{2+}$ ions and P$_2$W$_{16}$ clusters. Furthermore, the Ca$_2$-P$_2$W$_{16}$ NWs-based organogel shows fluorescence by exchanging organic molecules with fluorescent organic molecules (FOMs) in the organogel.

## Results and discussion
### Morphologies and structural characterizations of Ca$_2$-P$_2$W$_{16}$ NWs

It can be known from previous reports that the interaction between the organic molecules and ligands of NWs significantly impacts the performance of NWs-based organogels[15]. In this work, we designed and synthesized a highly charged POM cluster ([P$_2$W$_{18}$O$_{62}$]$^{6-}$) (Supplementary Fig. 1)[17,18]. Then, [Ca$_2$P$_2$W$_{16}$O$_{60}$]$^{10-}$ cluster units were obtained through the Ca$^{2+}$ embedded di-vacant [P$_2$W$_{16}$O$_{60}$]$^{14-}$ POM clusters. We fabricated the Ca$_2$-P$_2$W$_{16}$ NWs utilizing [Ca$_2$P$_2$W$_{16}$O$_{60}$]$^{10-}$ cluster units through a facile room-temperature reaction. As shown in Fig. 1c, the FTIR spectrum of the NWs exhibited characteristic absorption peaks of the Dawson-type POM cluster, indicating the presence of [P$_2$W$_{16}$O$_{60}$]$^{14-}$ POM clusters within NWs[19–21]. Absorption peaks at 2852 cm$^{-1}$ and 2918 cm$^{-1}$ belonged to oleylamine as identified in the FTIR spectra (Fig. 1c and Supplementary Fig. 6). We investigated the morphologies of Ca$_2$-P$_2$W$_{16}$ NWs through scanning transmission electron microscopy (STEM) and atomic force microscopy (AFM). The NWs were dispersed in octane and exhibited a curved and entangled structure with a length of several micrometers (Fig. 1a, b). The diameter of the NWs, approximately 1.7 nm, was determined from the AFM results (Fig. 1d, e), so the aspect ratio was as high as several thousand. The spacing distance between two NWs was calculated from the small-angle X-ray diffraction (SXRD), which was 4.0 nm (Fig. 1f). And their composition and structure were characterized using the inductively coupled plasma atomic emission spectrometry (ICP-AES). According to ICP-AES (Supplementary Table 1), the ratio of [P$_2$W$_{16}$O$_{60}$]$^{14-}$ to Ca$^{2+}$ in the NWs was 1:2. Therefore, the [Ca$_2$P$_2$W$_{16}$O$_{60}$]$^{10-}$ cluster units serve as building blocks for assembling NWs. Meanwhile, protonated oleylamine should be present on the NW to maintain the electrical neutrality of the NW.

As shown in the atomic-resolution AC high-angle annular-dark field scanning TEM (AC-HAADF-STEM) image (Fig. 1g), the NW was mainly composed of oval [Ca$_2$P$_2$W$_{16}$O$_{60}$]$^{10-}$ cluster units. The molecular model demonstrated that the Ca$^{2+}$ embedded [P$_2$W$_{16}$O$_{60}$]$^{14-}$ POM clusters in the stable state of NW (Fig. 1g and Supplementary Fig. 7). And oleylamine attached to NWs through coordination and electrostatic interactions (Fig. 1h). When exposed to ethanol, the Ca$_2$-P$_2$W$_{16}$ NWs tended to curl into nanocoils driven by hydrophobic forces, further proved their flexibility of polymer-like properties (Fig. 1i and Supplementary Fig. 8). The composition of the Ca$_2$-P$_2$W$_{16}$ NW was analyzed through X-ray photoelectron spectroscopy (XPS) (Supplementary Fig. 9), and energy-dispersive X-ray spectroscopy elemental

mapping (EDS). Ca$_2$-P$_2$W$_{16}$ NWs consisted of calcium (Ca), tungsten (W), phosphorus (P), and oxygen (O). The EDS mapping confirmed the uniform distribution of Ca, W, and P within the NWs coils (Fig. 1j).

### Single-molecule force spectroscopy results of NWs

In this part, we changed the flexibility of NWs by adjusting the type of POM cluster and the linked mode of units. Ca$_2$-P$_2$W$_{16}$ NWs, with a thickness of 1.7 nm, exhibit an inorganic skeleton while displaying characteristics reminiscent of carbon-carbon backbone polymers[22,23]. However, the flexibility of NW is mainly influenced by the flexibility of the inorganic skeleton and its surface chemical properties. Single-molecule force spectroscopy (SMFS) based on atomic force microscopy (AFM) has been a powerful tool for quantitatively investigating material mechanics at the nanoscale[24–26]. To quantitatively measure the PL of Ca$_2$-P$_2$W$_{16}$ NWs, we employed contact mode imaging on an Asylum Research Cypher VRS (Fig. 2a–c). The PL calculated by the worm-like chain (WLC) model corresponded to the rigidity of NWs. With a longer PL, the NW was more rigid[10]. The PLs of Ca$_2$-P$_2$W$_{16}$ NWs were mainly distributed (1.13 ± 0.31 nm) below that of single-stranded DNA (Fig. 2d)[27,28], suggesting that the flexible and free-rotating main strands form NWs that tend to bend and tangle to form gels in solution. However, the PL of Ca$_2$-P$_2$W$_{16}$ NW is higher than that of Ca-PW$_{12}$ NWs for two main reasons. One is that Ca$^{2+}$ acts as joints in the Ca$_2$-P$_2$W$_{16}$ NW, just like free-rotating carbon-carbon single bonds in organic polymers, thereby controlling the flexibility of the NW. However, the presence of two Ca$^{2+}$ ions at the joint partially restricts the flexibility of the NW. Additionally, the larger size of the Dawson-type POM cluster also means an increase in the length of the joint. Furthermore, we can calculate the rupture force of the NWs from the height of the peaks by analyzing these curves[29]. The rupture force of Ca$_2$-P$_2$W$_{16}$ (131.6 ± 21.7 pN, Fig. 2e) falls into the category of non-covalent interaction, indicating that the probe tip interacts with the NWs by van der Waals force rather than pulling the NW off.

In addition, transition-metal monosubstituted K$_x$P$_2$W$_{17}$MO$_{61}$ (M = TM, Fe/Mn/Ni/Co/Cr) POM clusters and rare earth metal monosubstituted K$_7$P$_2$W$_{17}$MO$_{61}$ (M = RE, Pr/Nd/Gd/Dy/Lu) POM clusters were synthesized (Supplementary Figs. 10a and 11a)[30,31]. And the corresponding Ca$_2$-P$_2$W$_{15}$M NWs were prepared (Supplementary Figs. 10b, 11b, 12, 13 and Supplementary Tables 2 and 3). In addition, the PLs of Ca$_2$-P$_2$W$_{15}$TM NWs (0.53–0.79 nm) and Ca$_2$-P$_2$W$_{15}$ RE NWs (0.93–1.21 nm) were mainly distributed above that of Ca-PW$_{12}$ NWs (Supplementary Figs. 14 and 15 and Supplementary Table 4). In a word, the flexibility of Ca$_2$-P$_2$W$_{16}$ and Ca$_2$-P$_2$W$_{15}$M NWs are generally lower than that of Ca-PW$_{12}$ NWs (Supplementary Fig. 16). These results supported the previous hypothesis (two Ca$^{2+}$ ions at the joint and the larger size of the [Ca$_2$P$_2$W$_{16}$O$_{60}$]$^{10-}$ POM cluster leading to the increase in PL). Then, we observed an increase in the rupture forces of Ca$_2$-P$_2$W$_{15}$M (M = RE) NWs (Supplementary Figs. 15 and 17). Thermal gravimetric analysis (TGA) proved that the increase in the rupture forces of Ca$_2$-P$_2$W$_{15}$M (M = RE) NWs may be attributed to the higher ligand density on the NW (Supplementary Fig. 18). Compared with Ca$_2$-P$_2$W$_{15}$Fe NWs, there are two more ligands on each POM surface in Ca$_2$-P$_2$W$_{15}$Pr NWs. This led to an increase in the van der Waals force between the NWs and the probe tip.

### Mechanical performances of NWs-based organogels

To understand the mechanical performances of the Ca$_2$-P$_2$W$_{16}$ NW-based organogel, a systematic investigation was conducted. First, upon standing at room temperature for 12 h, a freestanding Ca$_2$-P$_2$W$_{16}$ NW-based organogel can be obtained with a 10.0% mass fraction of Ca$_2$-P$_2$W$_{16}$ NWs (Ca$_2$-P$_2$W$_{16}$ NW-based organogel (10.0%)). These gels exhibit good elasticity and flexibility. When compressed or tensile strained, the gels can quickly regain their original shape, demonstrating their resilience (Fig. 3a, b and Supplementary Movies 1 and 2). Furthermore, the Ca$_2$-P$_2$W$_{16}$ NW-based organogel (10.0%) exhibits excellent stability, with a volume loss of 2.1% after 2 months of storage

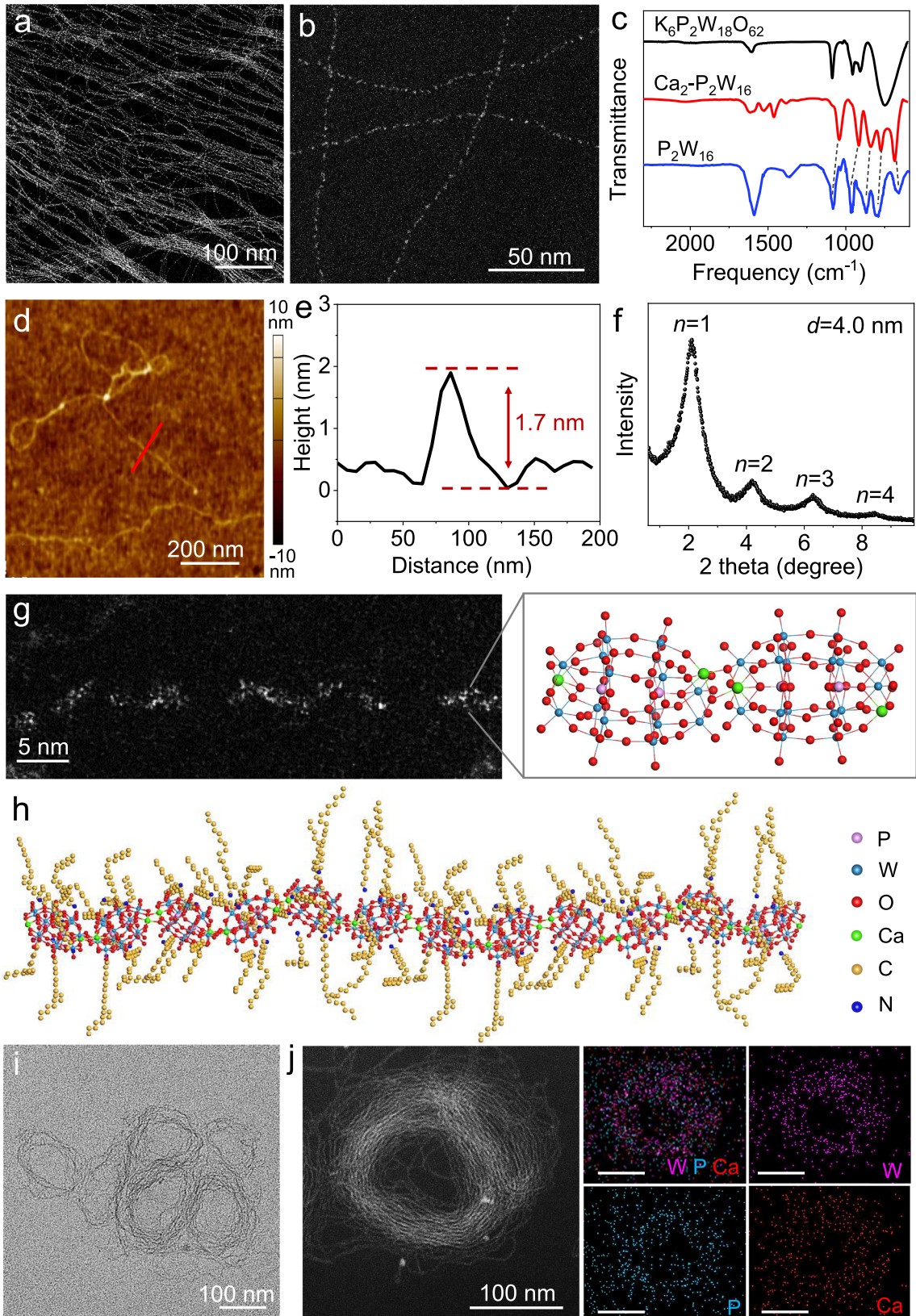

**Fig. 1 | Morphologies of Ca₂-P₂W₁₆ NWs. a**, **b** STEM images of Ca₂-P₂W₁₆ NWs. **c** FTIR spectroscopy of K₆P₂W₁₈O₆₂ (black), Ca₂-P₂W₁₆ NWs (red), and P₂W₁₆ POM (blue). **d** Typical AFM topographic image of Ca₂-P₂W₁₆ NWs. **e** The line profile along the dashed red line in (**d**). **f** SXRD pattern of Ca₂-P₂W₁₆ NWs. **g** AC-HAADF-STEM image, and molecular model of Ca₂-P₂W₁₆ NW structure. **h** The structural diagram of the NW. **i** TEM image of Ca₂-P₂W₁₆ coiled NWs. **j** STEM image and corresponding EDS elemental mapping images of Ca₂-P₂W₁₆ coiled NWs. Scale bars are 100 nm. Source data are provided as a Source Data file.

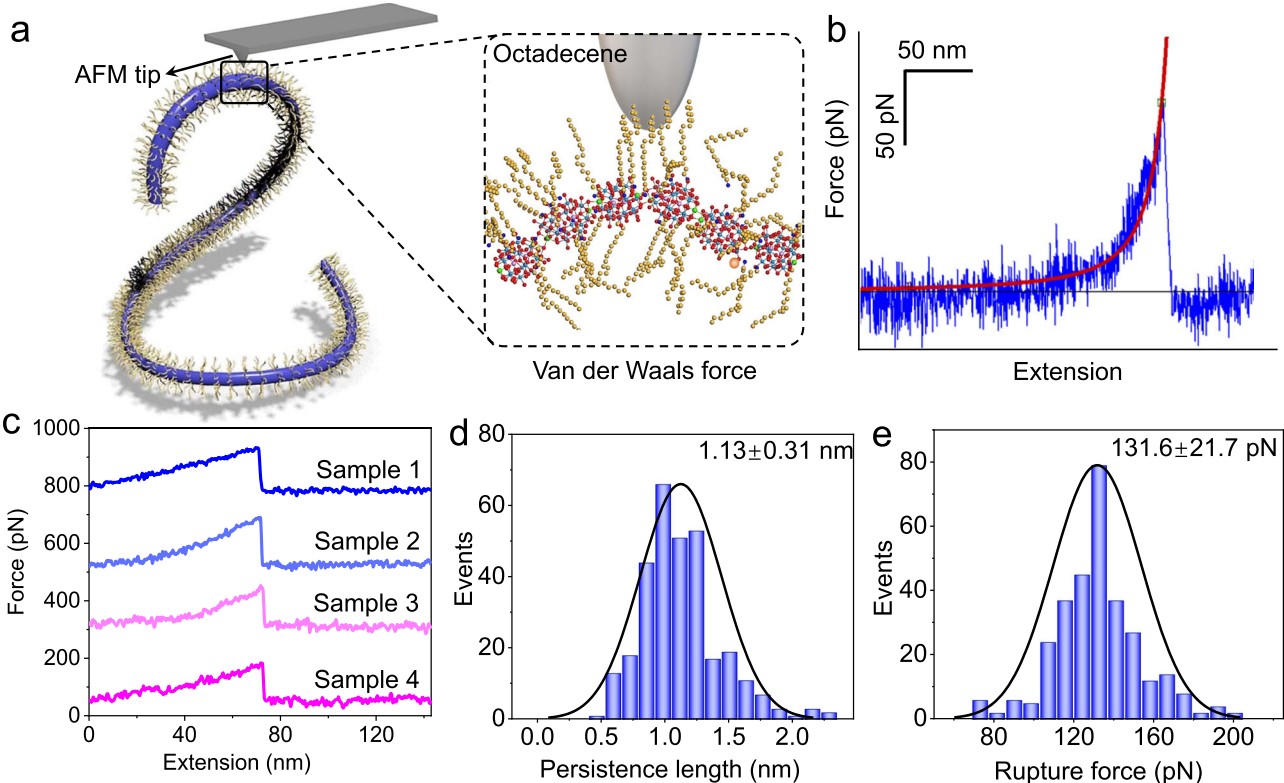

**Fig. 2 | Statistical results of SMFS tests. a** Schematic of SMFS test on $Ca_2$-$P_2W_{16}$ NWs. **b** A representative SMFS signal (blue) of $Ca_2$-$P_2W_{16}$ NWs, WLC model fitting curves (red). **c** Representative four samples (samples 1–4) of SMFS signals of $Ca_2$-$P_2W_{16}$ NWs. **d, e** Statistical diagrams of PL (**d**) and rupture force (**e**) of $Ca_2$-$P_2W_{16}$ NWs from 300 representative samples, respectively. Source data are provided as a Source Data file.

in a sealed container (Fig. 3c). The CGC varies for different organic liquids, owing to the distinct interactions between $Ca_2$-$P_2W_{16}$ NWs and solvent molecules (Fig. 3d, Supplementary Fig. 19 and Supplementary Table 5). We assume that the density of oleylamine coated on the NW is even distributed. Hence the density of oleylamine can be calculated to be ≈2 per Ca-$PW_{12}$ and ≈6 per $Ca_2$-$P_2W_{16}$ by TGA (Supplementary Fig. 4 and Supplementary Note 1).

The energy storage modulus (*G'*) of the $Ca_2$-$P_2W_{16}$-based organogel was higher than the loss modulus (*G''*), indicating that the gels have greater elasticity than viscosity. *G'* was higher than *G''* in the low angular velocity range and exhibited a rapid increase as ω increases, suggesting the presence of physical cross-link through entanglement (Fig. 3e). The mechanical properties of the $Ca_2$-$P_2W_{16}$ NW-based organogel with varying mass fraction of NWs were tested (Supplementary Fig. 20). In Fig. 3f, the compressive stress initially shows a slow and linear increase with strain, primarily due to the elastic deformation of the NWs network. The compressive stress of $Ca_2$-$P_2W_{16}$ NW-based organogel (10.0%) reaches up to 34.5 kPa at 60.4% strain. Thus, the compressive strength of the organogel is 0.0345 MPa. As shown in Fig. 3g, the elastic deformation of the NW network leads to the rapid and linear increase initially of tensile stress. However, the NW network gradually fractures when beyond the yield point (0.0273 MPa). The yield points of $Ca_2$-$P_2W_{16}$ NW-based organogels with different mass fractions are shown in Supplementary Table 6. The tensile stress of the $Ca_2$-$P_2W_{16}$ NW-based organogel (10.0%) reaches 29.0 kPa at the strain of 174.7%. The tensile strength of the organogel is up to 0.029 MPa. The mechanical performance of $Ca_2$-$P_2W_{16}$ NW-based gels represented improvements compared to Ca-$PW_{12}$ NW-based gels. In addition, the $Ca_2$-$P_2W_{16}$ NWs-octane gel boasts a low CGC of only 0.28% and exhibits very weak self-supporting and elastic properties. Supplementary Fig. 21 illustrates the results of the rheological, compressive stress-strain tests conducted on these gels. To assess the reproducibility and repeatability of

mechanical properties, specifically compression stress-strain and tensile stress-strain, we utilized $Ca_2$-$P_2W_{16}$ NW-based organogel (10.0%), using the same batch of NWs. These testing techniques represented good reproducibility and repeatability (Supplementary Fig. 22 and Supplementary Table 7). To estimate the interaction energy between the NWs and the solvent, we performed calculations (Supplementary Note 2)[32,33]. We assume that the alkyl chains (oleylamine) between NWs and the solvent (octane) aligned parallel to each other when approaching. Based on this simple model, we determined that the interaction energy between the NWs and the solvent can reach tens of $k_BT$, which is one to two orders of magnitude higher than the interaction energy between NWs. Considering the equation $\Gamma = \Gamma_0 + \Gamma_d$[34], where $\Gamma_0$ is the contribution from chemical cross-link (which is absent in the case of NW-based gel), the toughness of gels ($\Gamma$) mainly comes from $\Gamma_d$ (Supplementary Note 3). Therefore, the excellent mechanical properties of the gel stem not only from the entanglement of the NWs and the multi-stage interactions between them (driven by electrostatic and van der Waals forces). Strong interaction between the NWs and the organic liquid provides an important contribution. This is consistent with the previous MD simulations of Ca-$PW_{12}$ NW-based gels[9].

Under load-unload cycles, cracks may nucleate and propagate inside the gel, leading to changes in its elastic modulus, toughness, and other parameters. The fatigue behavior of the organogel is closely related to its rheological properties. Based on this, we propose that the fatigue mechanism of the organogel resembles viscoelastic fatigue and is reversible. The $Ca_2$-$P_2W_{16}$ NW-based organogel (10.0%) underwent 100 times compression-unload cycles test at 50.0% strain. After the first load, the compressive stress decreased by 18.2% and then gradually decreased. After 100 times of load-unload cycles, the maximum compressive-stress reduction is 53.3% (Fig. 3h, i). Similarly, the gel under the same conditions underwent a 100 times stretch-unload cycles test at 165.0% strain. The maximum tensile stress reduction

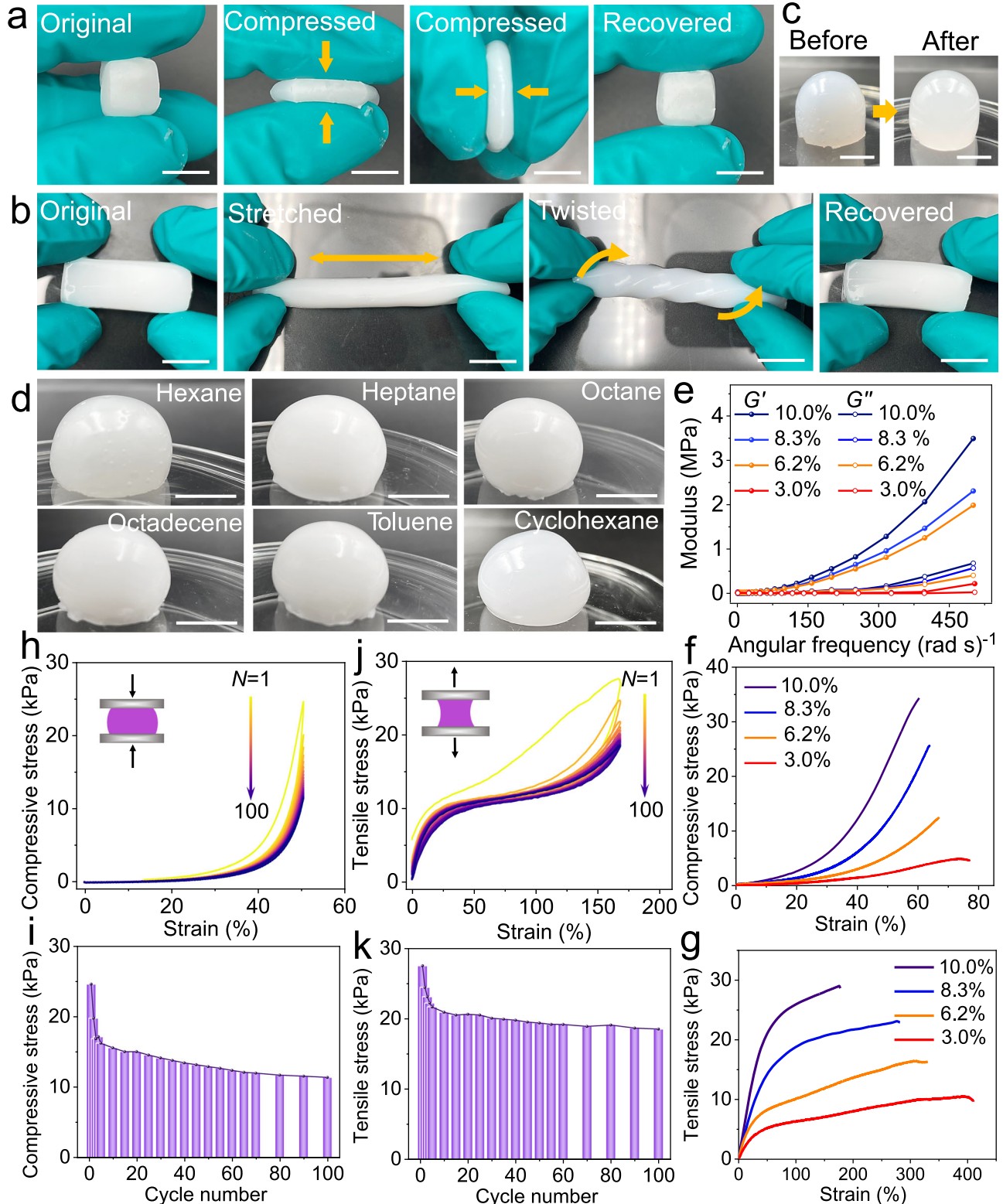

**Fig. 3 | Mechanical properties of Ca₂-P₂W₁₆ NW-octane gels. a**, **b** Photographs of the Ca₂-P₂W₁₆ NWs-octane gel, which was compressed and recovered (**a**) and was stretched, twisted, and recovered (**b**). **c** Photographs of the fresh Ca₂-P₂W₁₆ NW-octane gel volume loss of 2.1% after 2 months of storage in a sealed container. **d** Photographs of Ca₂-P₂W₁₆ NW-based gels with various organic liquids. **e** Rheological study of gels (10.0%: purple; 8.3%: blue; 6.2%: orange; 3.0%: red) in the frequency sweep mode for the strain amplitude of 1%. **f**, **g** Typical compressive stress-strain curves (**f**) and tensile stress-strain curves (**g**) of gels (10.0%: purple; 8.3%: blue; 6.2%: orange; 3.0%: red). **h**, **i** 100 cycles of compressive stress-strain curves of gels (10.0%) (**h**) and corresponding diagram of stress trend (**i**). **j**, **k** 100 cycles of tensile stress-strain curves of gels (10.0%) (**j**) and corresponding diagram of stress trend (**k**). Scale bars are 1 cm. Source data are provided as a Source Data file.

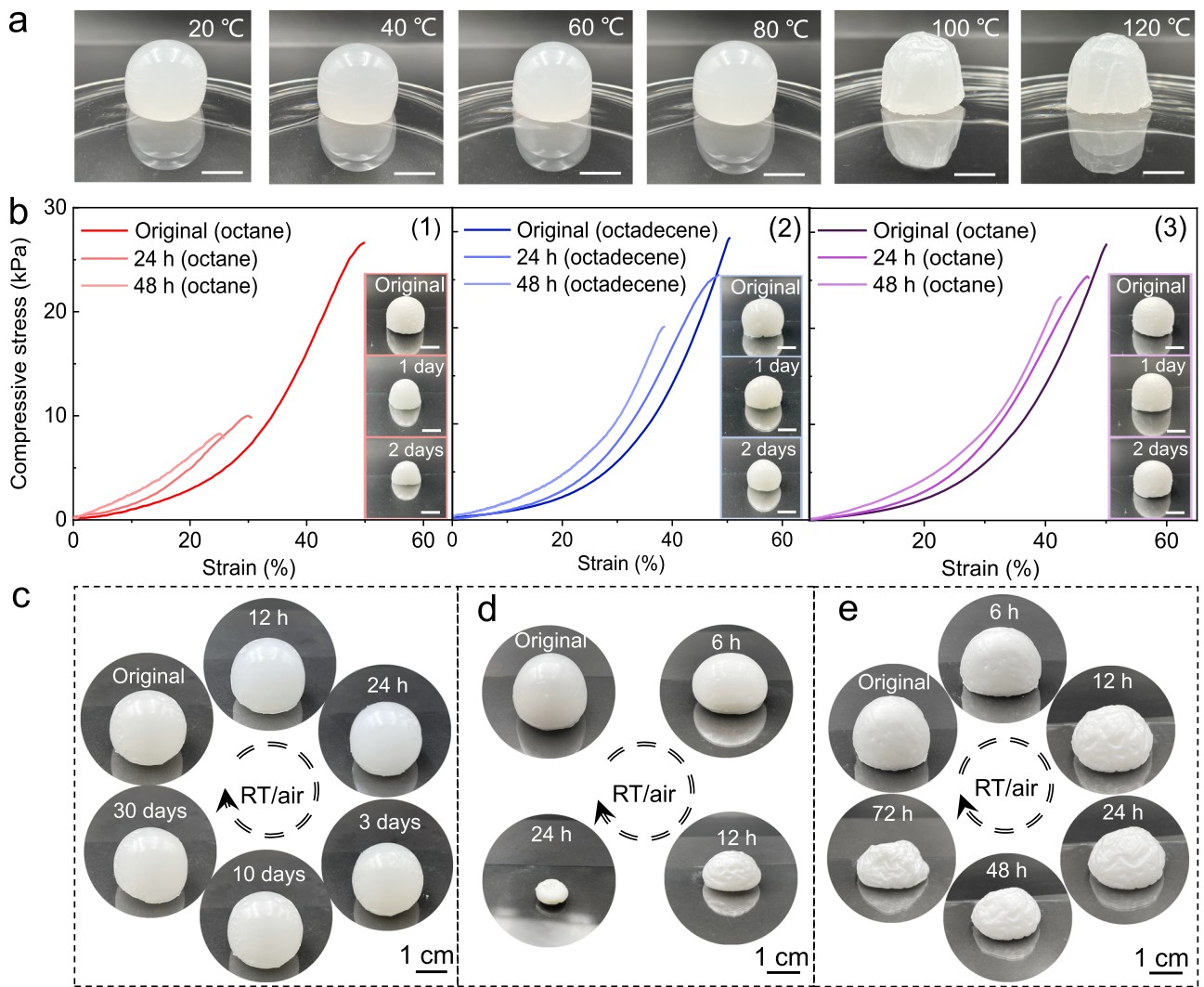

**Fig. 4 | Stability of Ca$_2$-P$_2$W$_{16}$ NW-based gels. a** Photographs of 10.0% Ca$_2$-P$_2$W$_{16}$ NW-octane gels stay at 20–120 °C for 30 min. **b** Typical compressive stress-strain curves of Ca$_2$-P$_2$W$_{16}$ NW-octane gels (1), Ca$_2$-P$_2$W$_{16}$ NW-octadecene gels (2), Ca$_2$-P$_2$W$_{16}$ NW-octane gels with NWs film (3) and insets are corresponding photographs of gels. **c–e** Photographs of a Ca$_2$-P$_2$W$_{16}$ NW-octadecene gel (**c**), Ca$_2$-P$_2$W$_{16}$ NW-octane gel (**d**), and Ca$_2$-P$_2$W$_{16}$ NW-octane gel with dense NW film on the surface (**e**) over time when placed in atmospheric condition at room temperature. The scale bars are 1 cm in (**a**) and (**b**). Source data are provided as a Source Data file.

reached to 32.8% (Fig. 3j, k). The experimental loading device is depicted in Supplementary Fig. 23. The results demonstrate that the gel exhibits excellent fatigue resistance. In addition to the Ca$_2$-P$_2$W$_{16}$ NW-octane gel, we also obtained Ca$_2$-P$_2$W$_{15}$M (M = Fe, Mn, Co, Cr) NW-octane organogels (Supplementary Fig. 24). The presence of Fe element in the coiled NWs can be observed in the EDS elemental mapping images (Supplementary Fig. 25). The Ca$_2$-P$_2$W$_{15}$M (M = Fe, Mn, Co, Cr) NWs-octane gels (10.0%) shows a compressive strain range of 50.1–54.2% and corresponding stress range of 28.8–31.8 kPa (Supplementary Fig. 26a). The tensile strain range is 124.9–168.8%, with corresponding stress range of 36.9–43.4 kPa (Supplementary Fig. 26b). Then, we prepared Ca$_2$-P$_2$W$_{15}$M (M = Pr, Nd, Gd, Dy, Lu) NW-based gels (Supplementary Fig. 27). A comparison of the compressive and tensile strains of these four NW-based gels (Ca-PW$_{12}$, Ca$_2$-P$_2$W$_{16}$ and Ca$_2$-P$_2$W$_{15}$M, 10.0%) demonstrated that mechanical properties represent an improvement (Supplementary Fig. 28). This improvement can be attributed to the stronger interaction between the Ca$_2$-P$_2$W$_{16}$ NWs and solvent molecules, as compared to Ca-PW$_{12}$ NWs.

Due to organogels having much broad range of liquid phase selectivity, their stability can be regulated by the boiling point of the solvent. We conducted stability tests on Ca$_2$-P$_2$W$_{16}$ NWs-based

organogels exposed to elevated temperature, aqueous environment, or atmospheric conditions. As shown in Fig. 4a, the Ca$_2$-P$_2$W$_{16}$ NW-octane gel remains stable in the temperature range of 20 to 80 °C. However, when the temperature reaches 100 °C, the organic solvent inside the gel begins to volatilize, resulting in the folding of the gel surface and shrinkage of the volume. As the temperature rose to 120 °C, the gel underwent more severe shrinkage and a large amount of solvent spilled out. However, the stability range of gels depended on the boiling point of the organic liquid. To improve the thermal stability of the gel, we dispersed the Ca$_2$-P$_2$W$_{16}$ NWs in high-boiling-point octadecene, which enhanced the thermal stability of the gel. The NW-octadecene gel remains stable without shrinkage or phase transition in the temperature range of 20 to 120 °C (Supplementary Fig. 29). In addition, the solvent in Ca$_2$-P$_2$W$_{16}$ NW-octane gel, exposed to atmospheric conditions at room temperature, volatilized only a small amount after one-month temperature, and the gel volume change was −5.1% (Fig. 4c). In Fig. 4b, Ca$_2$-P$_2$W$_{16}$ NW-octane gel and Ca$_2$-P$_2$W$_{16}$ NW-octadecene gel spill out partial solvent when they were immersed in water for 2 days (see (1) and (2) in Fig. 4b). Solvent loss leads to an increase in the mass fraction of NWs and a corresponding increase in compressive stress. However, the maximum strain was reduced by

49.9% and 24.1%, respectively. When the gel is immersed in water, the weak interaction between water and the organic molecules leads to a change in the structure of the gel and manifests as the contraction of the gel. In this process, $Ca_2$-$P_2W_{16}$ NW-based gel adapts to the new environment by releasing solvent molecules to reduce their volume[35]. To improve the stability of the gel in water, the gel was dried at 120 °C, and a dense NW film formed on the surface of NW-octane gel, which acts as a protective layer to slow down the rate of solvent release from the gel. In this case, the strain is reduced by only 14.8% after 2 days (see (3) in Fig. 4b). Based on these, we further investigated the protective effect of the dense NW film on internal solvents when the gel is exposed to atmospheric conditions at room temperature. As shown in Fig. 4d, the solvent in the NW-octane gel completely evaporates after 24 h, leaving only the NWs. However, even after 72 h, the octane has not completely evaporated, indicating a slower evaporation rate. Additionally, the gel gradually shrank over time (Fig. 4e).

### Fluorescence of $Ca_2$-$P_2W_{16}$ NWs-based organogels

Due to their distinctive properties, functionalized organogel hybrid materials offer promising applications in various fields. One example is that organogels were used to design light-harvesting assemblies[36]. Additionally, organogels have been employed for the stable formation and preservation of perovskite nanocrystals, resulting in excellent fluorescence efficiency. These properties have been extended to the development of soft electroluminescent and electronic devices[37–39]. However, enriching the functions of NW-based organogels is an opportunity with great challenges. In addition, the organic phase in polymer-based organogel exhibits dynamic characteristics rather than being static[40,41]. Based on this, we propose that $Ca_2$-$P_2W_{16}$ NWs-based gel can also exhibit similar dynamic properties to liquid solvents. Then, incorporating fluorescent dyes or fluorophores into an organogel and stimulating these dyes or groups will lead to the fluorescence of $Ca_2$-$P_2W_{16}$ NWs-based gels. The cured and freestanding gels were immersed in a 1 mM octane solution of fluorescent organic molecules (FOM/octane) (such as fluorescein, perylene diimide, and nile red) (Fig. 5a). Within 5 min, the FOMs gradually diffused into the gel (Fig. 5b–d). Within 5–20 min, the FOMs were evenly dispersed into the gel and formed aggregates. Then, the $Ca_2$-$P_2W_{16}$ NW-FOM/octane gels exhibited fluorescent properties, emitting green, orange, and red under an ultraviolet lamp ($\lambda = 365$ nm, 1.0 W cm$^{-2}$). The UV-vis spectra and fluorescence spectra showed the transition of these FOMs from monomer to aggregation, as evidenced by the broadening and redshift of the absorption peak (Fig. 5e–g and Supplementary Fig. 30)[42,43].

It is important that the $Ca_2$-$P_2W_{16}$ NW-FOM/octane gel remains well elasticity. The $Ca_2$-$P_2W_{16}$ NW-FOM/octane gel (10.0%) can rapidly recover its original shape after compression or tensile (Fig. 5h and Supplementary Movie 3). The FOMs can partially replace octane and interact with oleylamine through van der Waals force, as they are neutral molecules. The $Ca_2$-$P_2W_{16}$ NW-fluorescein/octane gel was soaked in octane for 12 h, during which some of the fluorescein molecules diffused out (Supplementary Fig. 31). As shown in Fig. 5i–k, fluorescein can be separated from the $Ca_2$-$P_2W_{16}$ NW-fluorescein/octane gel through evenly redistributing the gel in octane and then introducing ethanol. Furthermore, the $Ca_2$-$P_2W_{16}$ NW can be reassembled into a gel for reuse. Indeed, the utilization of organogel as a selective adsorbent for removing dye molecules has been achieved[44,45]. Therefore, NW-based organogels may be endowed with more functions through the introduction of other functional organic molecules, such as photosensitive, conductive, catalytic, and drug organic molecules, etc., or inorganic nanocrystals compatible with NWs. Based on the above results, more functional organogels can be developed and functional NW-based organogel systems will be constructed.

In summary, we prepared the $Ca_2$-$P_2W_{16}$ and $Ca_2$-$P_2W_{15}$M NWs through a facile room-temperature reaction, effectively trapping various volatile organic liquids. These NWs exhibit a low CGC, resulting in

improved mechanical properties and stability of the resulting organogels. Importantly, the PLs of $Ca_2$-$P_2W_{16}$ and $Ca_2$-$P_2W_{15}$M NWs were measured using single molecular force spectroscopy. Two $Ca^{2+}$ ions at the joint partially restrict the flexibility of the NWs, which proves the role of linked mode in tunning the flexibility of NWs. Incorporating fluorophores into the $Ca_2$-$P_2W_{16}$ NWs-based organogels realizes their fluorescence. In this study, high-performance, functional organogels were prepared with highly charged POM cluster units, and the connections between the microstructures of inorganic gelators and the properties of organogels were explored.

## Methods

### Materials

All chemicals were used as received without any further purification: $Ca(NO_3)_2 \cdot 4H_2O$ (99.0%), $H_3PW_{12}O_{40} \cdot xH_2O$ (PTA, AR), $Na_2WO_4 \cdot 2H_2O$ (99.5%), HCl (37%), $H_3PO_4$ (85%), KCl (99.8%), $MnCl_2 \cdot 4H_2O$ (99.9%), $Ni(NO_3)_2 \cdot 6H_2O$ (99.5%), $Co(NO_3)_2 \cdot 6H_2O$ (99.5%), $Cr(NO_3)_3 \cdot 9H_2O$ (99.0%), $PrCl_3 \cdot 6H_2O$ (99.5%), $Nd(NO_3)_3 \cdot 6H_2O$ (99.5%), $GdCl_3 \cdot 6H_2O$ (99.0%), $DyCl_3 \cdot 6H_2O$ (99.9%), $Lu(NO_3)_3 \cdot 6H_2O$ (99.5%), ethanol (99.5%), hexane (99.0%), heptane (99.5%), cyclohexane (99.5%), toluene (99.5%), chloroform (99.0%), absolute ethanol (99.5%), acetone (99.5%), formamide (99.5%), acetonitrile (99.0%) and ethylenediamine (99.0%) were purchased from Sinopharm Chemical Reagent Beijing Co., oleylamine (Sigma-Aldrich, 70%), octadecene (Sigma-Aldrich, 90%), fluorescein (Sigma-Aldrich, 98%), nile red (Sigma-Aldrich, 97.5%), perylene diimide (Aladdin, 95%). V-shaped $Si_3N_4$ AFM cantilevers (Bruker, MLCT, Beijing), silicon wafer (GinSRC, Beijing), carbon film coated grids (200 mesh) (EMCN, Beijing).

### Synthesis of $K_6P_2W_{18}O_{62} \cdot 14H_2O$[17,18]

$Na_2WO_4 \cdot 2H_2O$ (100 g, 0.3 mol) was dissolved in 120 mL deionized water. After the $Na_2WO_4 \cdot 2H_2O$ completely dissolved, HCl 4 M (83 mL) was added at a rate of 2 drops s$^{-1}$ with vigorous stirring. After the addition of HCl, the solution pH is between 6 and 7. Right away after the solution became clear and colorless, $H_3PO_4$ 4 M (83 mL) was added slowly at a rate of 4 drops s$^{-1}$. Upon completion of the phosphoric acid addition, the solution had a pH of 1–2. The solution was heated to boiling and refluxed for 24 h. Then, the resulting solution was cooled to room temperature, and KCl (50 g) was added to the solution, resulting in the formation of the precipitate. The precipitate was then filtered and air-dried. The crude material was dissolved in 200 mL of deionized water and heated to 80 °C. When small yellow crystals had formed, the solution was placed in a refrigerator set at 5 °C for recrystallization. The yellow crystals of $K_6P_2W_{18}O_{62} \cdot 14H_2O$ were collected by filtration and the corresponding MALDI-TOF-MS is shown in Supplementary Fig. 1.

### Synthesis of transition-metal monosubstituted $\alpha_2$-$K_xP_2W_{17}O_{61}M$ (M = TM, Fe/Mn/Ni/Co/Cr) and rare earth metal mono-substituted $\alpha_2$-$K_7P_2W_{15}O_{61}M$ (M = RE, Pr/Nd/Gd/Dy/Lu)[30,31]

General route to synthesize transition-metal monosubstituted $\alpha_2$-$K_xP_2W_{17}O_{61}M$ (M = TM): 5.2 g (1.1 mmol) of $a_2$-$K_{10}P_2W_{17}O_{61} \cdot 15H_2O$ was dissolved in 15 mL of 90 °C $H_2O$. A solution of 1.2 mmol of $Fe(NO_3)_3$ ($MnCl_2$, $Ni(NO_3)_2$, $Co(NO_3)_2$, $Cr(NO_3)_3$) in 4 mL of $H_2O$ was added with vigorous stirring, giving a deep brown solution. When the dissolution of the transition metal salt was complete, 164 mg (0.6 mmol) of $K_2S_2O_8$ in 2.5 mL of $H_2O$ was added. The solution was kept at 90 °C for 60 min. The oxidation is complete after 60 min. KCl (2 g) was added to the hot solution, and the solution was cooled to room temperature. The solution was then placed at 5 °C overnight. The resultant purple crystals were collected. Yields are falling between 70 and 90%.

General route to synthesize rare earth metal monosubstituted $\alpha_2$-$K_7P_2W_{17}O_{61}M$ (M = RE): $a_2$-$K_{10}P_2W_{17}O_{61} \cdot 15H_2O$ (5 g, 1.017 mmol) was dissolved in 50 mL of 0.5 M sodium acetate buffer at pH = 5.5 at 70 °C to form a clear solution. 3.08 mmol of $LnCl_3$ ($PrCl_3$, $Nd(NO_3)_3$, $GdCl_3$,

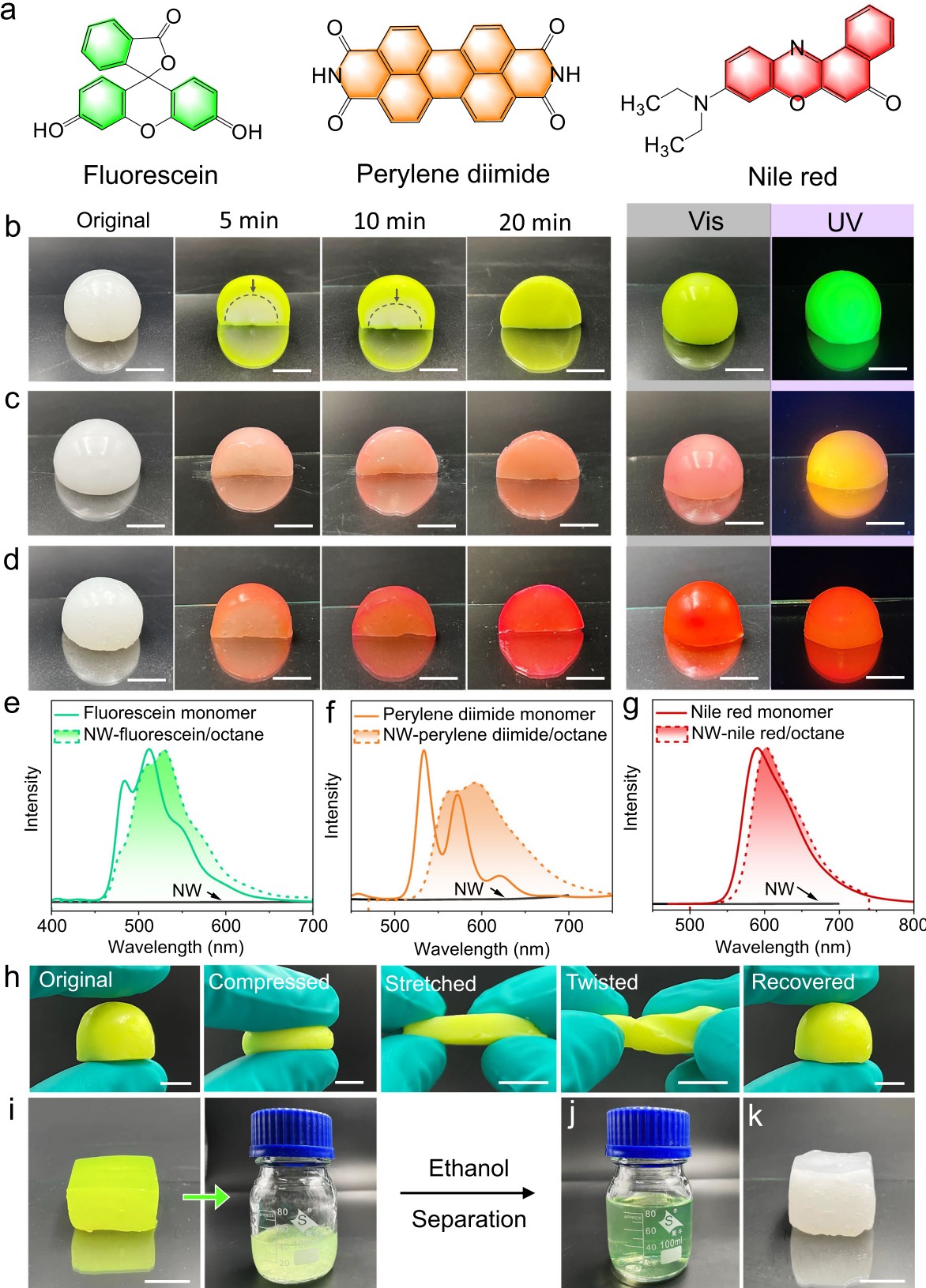

**Fig. 5 | Fluorescence of Ca₂-P₂W₁₆ NW-based gels. a** Chemical structures of FOMs (fluorescein, perylene diimide, and nile red). **b**–**d** Photographs of Ca₂-P₂W₁₆ NW-octane gels (10.0%) immersed in 1 mM of fluorescein/octane solution (**b**), perylene diimide/octane solution (**c**), and nile red/octane solution (**d**) over time. **e**–**g** The fluorescence spectra of Ca₂-P₂W₁₆ NW (black line), FOM monomers (color solid line), and Ca₂-P₂W₁₆ NW-FOM/octane gel (color dotted line). **h** Photographs of the Ca₂-P₂W₁₆ NW-fluorescein/octane gel, which was compressed, stretched, twisted, and recovered. **i**–**k** Photographs of the fluorescein recovery process with NWs. The gel was redispersed in the 40 mL octane (**i**). Added 40 mL ethanol to the mixture and centrifuged to obtain fluorescein solution as shown in (**j**) and NWs. The NWs were redispersed in the octane to form a gel for reuse (**k**). Scale bars are 1 cm. Source data are provided as a Source Data file.

$DyCl_3$, $Lu(NO_3)_3$) was dissolved in a minimum amount of water and added dropwise to the stirring $K_{10}P_2W_{17}O_{61}$ solution. KCl (5 g) was added to the reaction mixture, and the clear solution was cooled in the refrigerator. After 2 h, a white precipitate formed. Yields are falling between 65 and 90%.

The polyhedral representations of two monosubstituted Dawson-type POM isomers are shown in Supplementary Fig. 2.

### Synthesis of $Ca_2$-$P_2W_{16}$ and $Ca_2$-$P_2W_{15}M$ (M = Fe, Mn, Ni, Co, Cr, Pr, Nd, Gd, Dy, Lu) NWs[9]

1 g $K_6P_2W_{18}O_{62}$/$K_xP_2W_{17}MO_{61}$ and 0.08 g $Ca(NO_3)_2$·$4H_2O$ were dissolved in 16 mL deionized water. After stirring for 10 min, 12 mL of 1-octadecene and 4 mL of oleylamine were added in turn and it was stirred for 8 h. The product was washed and centrifuged at a speed of $1500 \times g$ three times using ethanol and octane, and $Ca_2$-$P_2W_{16}$ and $Ca_2$-$P_2W_{15}M$ NWs were obtained. Yields are falling between 73.3 and 59.6% (Supplementary Fig. 3).

### Preparation of the $Ca_2$-$P_2W_{16}$ NW-based and $Ca_2$-$P_2W_{15}M$ NW-based gel

$Ca_2$-$P_2W_{16}$ and $Ca_2$-$P_2W_{15}M$ NWs were dispersed in a proper volume of organic liquids well. The dispersion was centrifuged at a speed of $1500 \times g$ for 10 min to remove bubbles. The dispersion was stood for 8 h, and the gel was obtained.

### The method for evaluating mass percent of NWs in gel

NWs were dispersed in the volatile liquid, opening and standing, and the state of the dispersion was observed. When it gelled, a piece of gel was cut from the bulk and put into a centrifugal tube weighing $W_1$, and the total weight was recorded as $W_2$. Subsequently, put it in an oven and dried the gel at 60 °C for 8 h. The total weight of the centrifugal tube and the dried gel was recorded as $W_3$. The mass percent of NWs in the gel was $(W_3-W_1)/(W_2-W_1)*100\%$.

### Mechanical tests

The mechanical properties of gels were tested through the rheometer (Anton Paar, MCR301). Circular gel flakes with a diameter of 10 mm and thickness of 1 mm were used for rheological tests. The mechanical properties of gels under tension and pressure were characterized using a universal test machine (Instron 3400). The square gels with the size of $1 \times 1 \times 1$ cm$^3$ were used for compression tests and gel square flakes with the size of $2 \times 2 \times 0.5$ cm$^3$ were used for tension test. Both the tensile rate and compression rate were 10 mm min$^{-1}$.

### Procedures of SMFS test

The SMFS test was carried out on an Asylum Research Cypher VRS in contact mode. V-shaped $Si_3N_4$ AFM cantilevers (Bruker, MLCT) with a spring constant at ≈0.03 N m$^{-1}$ were used as received. The NWs were stretched and deflected by the AFM cantilever. Meanwhile, force-extension curves were recorded. The curves were then examined and fitted with the WLC model using Igor Pro, providing the persistence length. And each statistical result from 300 samples. In addition, the rupture forces of the NWs also obtained a statistical value by counting the peak values of 300 samples.

### Characterization techniques

Transmission electron microscope (TEM) images were obtained through the HITACHI H-7700 (100 kV). Energy-dispersive X-ray spectroscopy (EDS) elemental mapping results were obtained through the JEOL JEM (200 KV). Scanning electron microscope (SEM) images were obtained through the HITACHI SU8010. UV-vis absorption spectra were obtained using a UV-1900 spectrophotometer (Shimadzu). Fluorescence spectra were recorded using a FluoroMax-4 (Horiba). Atomic force microscope (AFM) results were

obtained through the Oxford instrument Cypher VRS. Atomic-resolution aberration-corrected TEM (AC TEM) JEOL-ARM200F (200 kV) was used to obtain high-angle annular-dark field scanning TEM (HAADF-STEM) images. Matrix-assisted laser desorption ionization time-of-flight mass spectrometry (MALDI-TOF-MS) analysis was performed on the MALDI-TOF/TOF-MS instrument (AXIMA Performance, Japan). Small-angle X-ray diffraction (SXRD) characterization was carried out on a Bruker D8 X-ray diffractometer using Cu Kα radiation ($\lambda = 1.5418$ Å). Inductively coupled plasma-atomic emission spectrometry (ICP-AES) was measured on the ThermoFisher iCAP 6300. X-ray photoelectron spectroscopy (XPS) tests were conducted on ULVAC-PHI Quantera SXM (operated at 250 kV, 55 eV with monochromated Al Kα radiation). Fourier-transform infrared (FTIR) spectra were obtained through a Nicolet 205 FTIR spectrometer.

### Reporting summary

Further information on research design is available in the Nature Portfolio Reporting Summary linked to this article.

## Data availability

The data supporting the findings of this study are available from the corresponding authors upon request. Source data are provided with this paper.

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

## Acknowledgements

This work was supported by NSFC (22241502, 22035004, 22250710677) (to X.W.), the China Postdoctoral Science Foundation (2023M731918 to F.Z. and 2022M721798 to Z.L.).

## Author contributions

X.W. contributed the idea and provided guidance. F.Z. performed most of the experiments. Z.L. provided theoretical analysis and supervision. W.X. wrote the manuscript with input from all authors.

## Competing interests

The authors declare no competing interests.
