## [Peer Review File · Nature Communications]

Mechanically tunable organogels from highly charged polyoxometalate clusters loaded with fluorescent dyesREVIEWER COMMENTS

Reviewer #1 (Remarks to the Author):
see attached pdf file.

Reviewer Comment for Manuscript ID: NCOMMS-23-31741

Manuscript Title: "Microstructurally-regulated organogels properties mediated by highly charged polyoxometalates cluster"

The authors prepared an organogel from $\text{Ca}_2\text{-P}_2\text{W}_{16}$ nanowires. The organogel exhibited robust mechanical strength and its flexibility is commendable. The manuscript exhibits several strong points. The experimental design is well-structured, and the methodology, characterization and interpretation of the results are sound. The supplementary videos are very convincing. The references are relevant and updated. However, a few concerns must be addressed before the manuscript can be accepted.

The current title of the manuscript is vague. The authors should provide a clear and concise title.

$\text{Ca}_2\text{-P}_2\text{W}_{16}$ is introduced for the first time in the abstract and the full name must be provided.

It is mentioned in the abstract that only 0.28% mass fraction of the $\text{Ca}_2\text{-P}_2\text{W}_{16}$ nanowires were used to construct the gel. However, there is no discussion on this mass fraction in the remainder of the manuscript. Further, the rheological, compressive, and tensile stress studies show the minimum mass fraction used is 3%. The authors should clarify this.

Line 96, remove the sentence and associated table and graph, Sr-based SNWs are only mentioned in this instance.

Line 124-128, sentence incomplete.

Did the authors calculate the yield point of the gel and the gel strength? This has to be provided in the manuscript.

What is the purpose of FOMs in this study? Are they used to prove that organogels can trap molecules? Also, what is the kind of interaction between the gel and the FOMs? What happens to the FOMs after washing the gel?

Additionally, there are a few typographical errors in the manuscript (e.g., lines 67 and 79 "becomed" should be corrected to "became" and in line 77 space between the words "atomicresolution", line 122 "bellowing should be changed to "below"). Line 36, the sentence must be reworded.

The authors can include a few recent studies on organogels (e.g., 10.1038/s41467-020-18383-y, 10.1038/ncomms7650, 10.1021/acs.langmuir.1c00288, 10.1016/j.matlet.2020.127854, 10.1021/acsami.9b21697) which are relevant to the study presented.

Supplementary Information:

Rewrite line 35.

Supplementary Fig.1 given doesn't correlate with the text provided on page 3, line 49

Line 53, reference needed.

1.3 is tough to follow and needs to be simplified.

Line 55, a mixture of a1 and a2 products, what is the difference between a1 and a2 and how are they distinguished?

1.4, in the replacement of K by Ca forming $\text{Ca}_2\text{-P}_2\text{W}_{16}$, what is the role of octadecene and oleylamine in nanowire formation?

1.4, How much is the yield of $\text{Ca}_2\text{-P}_2\text{W}_{16}$?

Line 77, temperature set in the oven needs to be provided.

Line 81, Sentence incomplete.

Line 153, 162, Reference needed.

Line 152 and 160, Paralleled should be changed to parallel.

Overall, I find this manuscript has a significant impact on the scientific community. However, the authors must address the concerns raised and therefore, I recommend "a minor revision".

Reviewer #2 (Remarks to the Author):

After a careful reading of the manuscript, this research article is recommended for publication in Nature Communications, with some slight remarks and suggestions.

- The statistical analysis is not really explained. Please be more precise, show results and discuss them.
- Please be more precise about the reproducibility and repeatability of each technique used.
- Please include the sources of all materials used in the study.
- References should be checked as some references are not properly formatted.

Reviewer #3 (Remarks to the Author):

In this work, a new class of molecular gelator for organic liquids is reported - i.e. subnanometer inorganic nanowires formed by polyoxometalates clusters, linked by calcium ions. This work explores how these nanowires are formed, at the molecular level, using combinations of electron microscopy and spectroscopy, molecular simulations, and mechanical characterization. It also expands the available library of gelators by substituting W atoms by Fe or Pr, for example. Three main results stand out: (1) the capacity to probe the strength of fibers at the nm level using AFM; (2) the very high quality microscopy analysis coupled with the molecular simulations, offering a clear perspective on how these fibers are formed; (3) the significant mechanical properties that the gel displays - most notably their elasticity.

In my opinion, the experimental work is of very high quality, and the molecular simulations truly bring additional elements of comprehension. However, I do have some reserve at this point regarding the novelty of the results, their importance for a large scientific audience, and their impact. The gelator itself that is the object of this work - mostly $\text{Ca}_2\text{-P}_2\text{W}_{16}$ - appears to have been reported previously by the authors in Ref.9. In this work, a number of features, including the fiber-like structure and mechanical properties, are presented. It is not clear to me then how the results in the current submitted work differ or expand on those presented in ref.9, or bring truly new insight.

The results in Figure 2, about the Single-molecule force spectroscopy, also seem to expand on those presented in ref.10 by the same authors. It is not clear how the results presented here really bring new original insight, or if it is a refinement.

My understanding at this point then is that the new content is really about tuning the properties of the gelator and resulting gel by substituting W atoms by transition metals (herein Fe, for example) or rare earths (RE). It seems however that the main figures are mostly concerned with the basic gelator, whereas the results with the substituted gelators are presented as Supporting information.

I am not convinced at this point then that the content fits the scope of Nature Communications - it would be better suited I believe for a high quality more specialized journal. In addition, the beginning of the Introduction is quite generic, and it is not clear at all why this work is important for the broad readership of Nature Communications - needs to be better explained and put in context. However, I would be open to consider the arguments of the authors, and to reconsider my recommendation if need be.

Response to comments from Reviewer #1

Manuscript Title: “Microstructurally-regulated organogels properties mediated by highly charged polyoxometalates cluster” The authors prepared an organogel from $\text{Ca}_2\text{-P}_2\text{W}_{16}$ nanowires. The organogel exhibited robust mechanical strength and its flexibility is commendable. The manuscript exhibits several strong points. The experimental design is well-structured, and the methodology, characterization and interpretation of the results are sound. The supplementary videos are very convincing. The references are relevant and updated. However, a few concerns must be addressed before the manuscript can be accepted.

Reply: We thank reviewer #1 for the nice comments, and hope that our revision (see below) will clarify his/her concerns.

Comment 1. The current title of the manuscript is vague. The authors should provide a clear and concise title. $\text{Ca}_2\text{-P}_2\text{W}_{16}$ is introduced for the first time in the abstract and the full name must be provided. It is mentioned in the abstract that only 0.28% mass fraction of the $\text{Ca}_2\text{-P}_2\text{W}_{16}$ nanowires were used to construct the gel. However, there is no discussion on this mass fraction in the remainder of the manuscript. Further, the rheological, compressive, and tensile stress studies show the minimum mass fraction used is 3%. The authors should clarify this.

Reply: We sincerely thank this reviewer for pointing this issue out.

(1) The word “microstructurally-regulated” is not clear and concise in the original title, so it has been deleted in the revised manuscript. The “Highly charged polyoxometalates cluster” represents a novel kind of building block characterized by their highly negative charge (P_2W_{16} and $\text{P}_2\text{W}_{15}\text{M}$ cluster with the negative charges above 10), while the charge of the PW_{12} cluster unit in Ref. 9 is only 3. Consequently, we revised the title of the article to “**Sub-nanowires formed by highly charged polyoxometalate clusters for improving mechanical properties of organogels**” Previous studies (*Science* 2022, **377**, 100-104) have emphasized the crucial role of interaction forces between solvent

molecules and oleylamine in the SNWs gelation process. A subsequent review (*ACS Nano* 2023, **17**, 20-26) presented that increasing the ligand density on the SNW surface effectively can strengthen the interaction between solvent molecules and oleylamine. To achieve this, we used the higher charge POM clusters as building blocks to form SNWs, resulting in SNWs with higher ligand density and SNW-based organogels with better macroscopic mechanical properties. This study is to enhance the mechanical properties of organic gels on a macroscopic scale by improving the solvent-locking abilities of SNW-based organic gels and endowing organogels with optical properties, highlighting the hierarchical relationship among “highly charged POM clusters”, “SNWs with high ligand density” and the resulting “improved mechanical properties” of SNW-based organogels.

(2) We have provided the full name of $\text{Ca}_2\text{-P}_2\text{W}_{16}$ in the abstract of the revised manuscript as follows:

“Here, we fabricated the $\text{Ca}_2\text{-P}_2\text{W}_{16}$ and $\text{Ca}_2\text{-P}_2\text{W}_{15}\text{M}$ SNWs utilizing highly charged $[\text{Ca}_2\text{P}_2\text{W}_{16}\text{O}_{60}]^{10-}$ and $[\text{Ca}_2\text{P}_2\text{W}_{15}\text{MO}_{60}]^{14-/13-}$ cluster units, respectively, which were then employed for preparing organogels.”

(3) **The Critical Gel Concentration (CGC) of a gel is the lowest concentration of gel-forming agents required to induce gelation in the corresponding solvent.** Gels formed at this concentration exhibit very weak self-supporting and elastic properties. In this work, the CGC of $\text{Ca}_2\text{-P}_2\text{W}_{16}$ SNWs-based gel is 0.28%. Photographs in **Fig. R1a** of the $\text{Ca}_2\text{-P}_2\text{W}_{16}$ SNWs-based gel, clearly demonstrate its free-standing and elastic properties. **Fig. R1b** illustrates the results of the rheological, compressive stress-strain tests conducted on these gels. In the revised manuscript, this part has been included in **Supplementary Fig. 21.**

However, 0.28% is a limit mass fraction of $\text{Ca}_2\text{-P}_2\text{W}_{16}$ SNWs in octane, which serves as a parameter indicative of the gelation performance of these $\text{Ca}_2\text{-P}_2\text{W}_{16}$ SNWs. Operating the $\text{Ca}_2\text{-P}_2\text{W}_{16}$ SNW-based organogels (0.28%) during mechanical properties tests presents significant challenges, with poor reproducibility and repeatability of each technique used. **Consequently, the mechanical strength of the organogels (0.28%)**

lacks representativeness, leading to the mechanical strength data of the gels not appearing in the manuscript.

To explore the relationship between the mass fraction of SNW and the mechanical properties of organogels, we conducted a comparative analysis involving four different mass fractions: 3.0%, 6.2%, 8.3%, and 10.0%. To compare test results in various aspects, we used 3% to be the minimum mass fraction for rheological, compressive, and tensile stress studies.

Fig. R1. Mechanical properties of 0.28% $\text{Ca}_2\text{-P}_2\text{W}_{16}$ SNW-octane gels. **a**, Photographs of the 0.28% $\text{Ca}_2\text{-P}_2\text{W}_{16}$ SNWs-octane gel were compressed and recovered. **b**, Typical compressive stress-strain curves of 0.28% $\text{Ca}_2\text{-P}_2\text{W}_{16}$ SNW-octane gels. Element symbols and numbers in the color keys indicate four gel samples with the same mass fraction. **c**, Rheological study of gels in the frequency sweep mode for the strain amplitude of 1%.

Comment 2. Line 96, remove the sentence and associated table and graph, Sr-based SNWs are only mentioned in this instance.

Reply: Thanks for the comment. We have removed the sentence and associated table and graph of Sr-based SNWs in the revised manuscript and the revised supplementary

information.

Comment 3. Line 124-128, sentence incomplete.

Reply: Thanks for the comment. We have done a more thorough introduction on the connected mode and internal structure of SNWs, which were added to the revised manuscript as follows:

“Thermal gravimetric analysis (TGA) proved that the increase in the rupture forces of $\text{Ca}_2\text{-P}_2\text{W}_{15}\text{M}$ (M=RE) SNWs may be attributed to the higher ligand density on the SNW (**Supplementary Fig. 18**). Compared with $\text{Ca}_2\text{-P}_2\text{W}_{15}\text{Fe}$ SNWs, there are two more ligands on each POM surface in $\text{Ca}_2\text{-P}_2\text{W}_{15}\text{Pr}$ SNWs. This led to an increase in the van der Waals force between the SNWs and the probe tip. In a word, we can change the flexibility of SNWs by adjusting the type of POM cluster and the linked mode of units.”

Comment 4. Did the authors calculate the yield point of the gel and the gel strength? This has to be provided in the manuscript.

Reply: Thanks for the suggestion. We will address the reviewer's comments by focusing on two key aspects:

(1) Gels are typically viscoelastic materials, which means they exhibit both solid-like (elastic) and liquid-like (viscous) properties. Below the yield point, the gel behaves like a solid, and it can maintain its shape and resist deformation. However, when a stress or force is applied that exceeds the yield point, the gel will start to flow or deform like a liquid, and it may not fully return to its original shape when the stress is removed. This behavior is often described as "yielding" or "yield stress" in the context of gels.

In the case of SNW-based organogels, stress initially exhibited a rapid and linear increase with strain, primarily attributed to the elastic deformation of the SNW networks. Beyond the yield point, the SNW networks progressively fractured under tension, resulting in permanent deformation of the gels. Ultimately, a crack developed in the gel, propagating under tension until the gel ultimately ruptured. With this understanding in mind, we calculated the yield points of SNW-based organogels with varying solid contents (10.0%,

8.3%, 6.2%, 3.0%). The results of this analysis are presented in **Table R1**, which has also been included in the revised **Supplementary Table 5**. The yield point and the gel strength are added in the revised manuscript.

Table R1. The yield points of SNW-based organogels

Samples		Mass fraction (%)	Yield point (MPa)
Ca₂-P₂W₁₆	Ca ₂ -P ₂ W ₁₆	10.0%	0.0273
	Ca ₂ -P ₂ W ₁₆	8.3%	0.0196
	Ca ₂ -P ₂ W ₁₆	6.2%	0.0122
	Ca ₂ -P ₂ W ₁₆	3.0%	0.0071
Ca₂-P₂W₁₅M (M=TM)	Ca ₂ -P ₂ W ₁₅ Fe	10.0%	0.0264
	Ca ₂ -P ₂ W ₁₅ Mn	10.0%	0.0231
	Ca ₂ -P ₂ W ₁₅ Co	10.0%	0.0191
	Ca ₂ -P ₂ W ₁₅ Ni	10.0%	0.0246
	Ca ₂ -P ₂ W ₁₅ Cr	10.0%	0.0241
Ca₂-P₂W₁₅M (M=RE)	Ca ₂ -P ₂ W ₁₅ Pr	10.0%	0.0244
	Ca ₂ -P ₂ W ₁₅ Nd	10.0%	0.0232
	Ca ₂ -P ₂ W ₁₅ Gd	10.0%	0.0229
	Ca ₂ -P ₂ W ₁₅ Dy	10.0%	0.0231
	Ca ₂ -P ₂ W ₁₅ Lu	10.0%	0.0256

(2) We calculate the tensile strength and compressive strength of the gels by considering the ratio of the stress at the curve's inflection point when the gel breaks down to the force

area. The calculation results are presented in **Table R2**, which has also been included in the revised **Supplementary Table 6**.

Table R2. The tensile strength and compressive strength of Ca₂-P₂W₁₆ SNW-based organogels

Samples	Mass fraction (%)	Tensile strength (kPa)	Compressive strength (kPa)
Ca ₂ -P ₂ W ₁₆	10.0%	29.0	34.5
Ca ₂ -P ₂ W ₁₆	8.3%	23.1	25.5
Ca ₂ -P ₂ W ₁₆	6.2%	16.6	12.5
Ca ₂ -P ₂ W ₁₆	3.0%	10.5	4.9

Comment 5. What is the purpose of FOMs in this study? Are they used to prove that organogels can trap molecules? Also, what is the kind of interaction between the gel and the FOMs? What happens to the FOMs after washing the gel?

Reply: Thanks for the comment. The organic phase in polymer-based organogel exhibits dynamic characteristics rather than being static. (*Langmuir* 2015, **31**, 13850-13859; *J. Control. Release*. 2018, **271**, 1-20.) We propose that SNWs-based gel can also exhibit similar dynamic properties to liquid solvents. In addition, incorporating fluorescent dyes or fluorophores into an organogel and stimulating these dyes or groups to lead the photoluminescence of SNWs-based gels. (*Chem. Soc. Rev.* 2008, **37**, 109-122; *Angew. Chem. Int. Ed.* 2012, **51**, 1760-1762; *Langmuir* 2021, **37**, 3996-4006).

(1) In this work, our purpose is threefold. Firstly, we opted for neutral fluorescent molecules. This allowed the FOMs to partially substitute for octane and engage with oleylamine through van der Waals forces. Secondly, fluorescent molecules have the advantage of displaying color, enabling us to observe the diffusion of molecules within the gel without compromising the integrity of the SNWs or the gel structure. Thirdly, owing to the relatively weak van der Waals force, the FOMs can be eventually separated from the gel, allowing

for the recycling of both the FOMs and the SNWs (**Fig. R2f-h**). This part has been included in **Fig. 5i-k** of the revised manuscript.

(2) Proving that organogels can trap molecules is indeed one of our purposes, but it is not all. The current experimental results demonstrate the effective capture and release of small molecules of neutral organic fluorescence by the gel under specific conditions. However, the capability of capturing charged molecules and larger molecules remains uncertain and requires further investigation, which will be the focus of our next research endeavor.

(3) The FOMs can interact with oleylamine through van der Waals force, as they are neutral molecules. The $\text{Ca}_2\text{-P}_2\text{W}_{16}$ SNW-fluorescein/octane gel was soaked in octane for 12 hours, during which some of the fluorescein molecules diffused out. This also confirms that the interaction between FOM and oleylamine is primarily a weak van der Waals force (**Fig. R2a-e**). This part has been added in the revised **Supplementary Fig. 31**.

Fig. R2. **a**, Photograph of $\text{Ca}_2\text{-P}_2\text{W}_{16}$ SNW-fluorescein/octane gel (10%). **b-d**, Photographs of the gels soaked in octane for 2, 6, and 12 hours. **e**, Photograph of the gel soaked in octane for 12 hours. **f-h**, Photographs of the fluorescein recovery process with SNWs. The gel was redispersed in the 40 mL octane (**f**). Added 40 mL ethanol to the mixture and centrifuged to obtain fluorescein solution as shown in (**g**) and SNWs. The SNWs were redispersed in the octane to form a gel for reuse (**h**). Scale bars are 1 cm.

(4) As shown in **Fig. R2. a-e**, after washing the $\text{Ca}_2\text{-P}_2\text{W}_{16}$ SNW-fluorescein/octane gel, some of the fluorescein molecules diffused out. However, As shown in **Fig. R2f**, fluorescein can be separated from the $\text{Ca}_2\text{-P}_2\text{W}_{16}$ SNW-fluorescein/octane gel by evenly redistributing the gel in octane and introducing ethanol. Furthermore, the $\text{Ca}_2\text{-P}_2\text{W}_{16}$ SNW can be reassembled into a gel for reuse (**Fig. R2f-h**).

Comment 6. Additionally, there are a few typographical errors in the manuscript (e.g., lines 67 and 79 "becomed" should be corrected to "became" and in line 77 space between the words "atomicresolution", line 122 "bellowing should be changed to "below"). Line 36, the sentence must be reworded.

Reply: Thanks for the suggestions.

(1) Concerning the reviewer's comments, we have made corresponding revisions in the revised manuscript.

Line 67: This line has been deleted in the revised manuscript.

Line 77 and Line 79: "As shown in the atomic-resolution AC high-angle annular-dark field scanning TEM (AC-HAADF-STEM) image, the SNW was mainly composed of oval $[\text{Ca}_2\text{P}_2\text{W}_{16}\text{O}_{60}]^{10-}$ cluster units."

Line 122: "In addition, the PLs of $\text{Ca}_2\text{-P}_2\text{W}_{15}\text{Fe}$ SNWs (0.58 ± 0.28 nm) and $\text{Ca}_2\text{-P}_2\text{W}_{15}\text{Pr}$ SNWs (0.93 ± 0.19 nm) were mainly distributed above that of Ca-PW_{12} SNWs (**Supplementary Figs. 15, 16**)."

(2) We have reworded the sentence in the introduction of the revised manuscript as follows:

Line 36: "AE- PW_{12} SNWs are assembled by the electrostatic interactions of alkaline earth metal cations and $[\text{PW}_{12}\text{O}_{40}]^{3-}$ POM clusters, which can easily form 3D interwoven structures through electrostatic and van der Waals interactions, efficiently trapping organic solvents for easy recycling."

Comment 7. The authors can include a few recent studies on organogels (e.g., 10.1038/s41467-020-18383-y, 10.1038/ncomms7650, 10.1021/acs.langmuir.1c00288, 10.1016/j.matlet.2020.127854, 10.1021/acsami.9b21697) which are relevant to the

study presented.

Reply: Thanks for the comment. We have done a more thorough literature review on the functionality of organogels, which was summarized, and the literature was cited in the discussion section “*Photoluminescence of Ca₂-P₂W₁₆ SNWs-based gels*” of the revised manuscript. We have cited the aforementioned literature and rationalized the claim in the introduction section as follows:

“Functionalized organogel hybrid materials offer promising applications in various fields, owing to their distinctive properties. One example is that organogels were used to design light-harvesting assemblies.³⁶ Additionally, organogels have been employed for the stable formation and preservation of perovskite nanocrystals, resulting in excellent photoluminescence efficiency. These properties have been extended to the development of soft electroluminescent and electronic devices.³⁷⁻³⁹”

“Indeed, the utilization of organogel as a selective adsorbent for removing dye molecules has been achieved.^{44,45} Therefore, SNW-based organogels may be endowed with more functions through the introduction of other functional organic molecules, such as photosensitive, conductive, catalytic, and drug organic molecules, etc., or inorganic nanocrystals compatible with SNWs. Based on the above results, more functional organogels can be developed and functional SNW-based organogel systems will be constructed.”

Supplementary Information:

Comment 8. Rewrite line 35.

Reply: Thanks for the comment. We have rewritten this part in the revised supplementary information as follows:

Line 35: “Right away after the solution became clear and colorless, H₃PO₄ 4 M (83 mL) was added slowly at a rate of 4 drops/second. Upon completion of the phosphoric acid addition, the solution was clear, exhibited a pale-yellow color, and a pH of 1-2”

Comment 9. Supplementary Fig.1 given doesn't correlate with the text provided on page

3, line 49. Line 53, reference needed.

Reply: Thanks for the comments. In response to the reviewer's comments, we have completed the following two changes.

(1) The introduction to **Supplementary Fig. 1** in the original supplementary information is not clear enough. **Supplementary Fig. 1** corresponds to the Matrix-assisted laser desorption ionization time-of-flight mass spectrometry (MALDI-TOF-MS) of product $K_6P_2W_{18}O_{62} \cdot 14H_2O$, which we have modified in the revised supplementary information, as shown below:

Line 49: "The yellow crystals of $K_6P_2W_{18}O_{62} \cdot 14H_2O$ were collected by filtration and corresponding Matrix-assisted laser desorption ionization time-of-flight mass spectrometry (MALDI-TOF-MS) is shown in **Supplementary Fig. 1**"

(2) We have rewritten this part in the revised supplementary information and included proper references as follows:

Line 53: "General route to synthesize transition-metal monosubstituted $\alpha_2-K_xP_2W_{17}O_{61}M$ ($M=TM, Fe/Mn/Ni/Co/Cr$): These compounds were prepared by an adaptation of synthesis routes described in the literature.^{3,4} (*J. Am. Chem. Soc.* 1991, **113**, 7209-7221; *J. Chem. Soc. A* 1968, **0**, 2647.)"

These two literatures are included in the revised manuscript as references 28 and 29.

Comment 10. 1.3 is tough to follow and needs to be simplified.

Reply: Thanks for the comment. The synthesis method of 1.3 lacks sufficient detail. We have rewritten the revised supplementary information and included proper references as follows:

"1.3 Synthesis of transition-metal monosubstituted $\alpha_2-K_xP_2W_{17}O_{61}M$ ($M=TM, Fe/Mn/Ni/Co/Cr$) and rare earth metal monosubstituted $\alpha_2-K_7P_2W_{15}O_{61}M$ ($M=RE, Pr/Nd/Gd/Dy/Lu$)^{3,4}

(1) General route to synthesize transition-metal monosubstituted $\alpha_2-K_xP_2W_{17}O_{61}M$ ($M=TM, Fe/Mn/Ni/Co/Cr$): These compounds were prepared by an adaptation of synthesis routes

described in the literature.^{3,4} In a 50 mL flask, 5.2 g (1.1 mmol) of α_2 -K₁₀P₂W₁₇O₆₁·15H₂O was dissolved in 15 mL of 90°C H₂O. A solution of 1.2 mmol of Fe(NO₃)₃ (MnCl₂, Ni(NO₃)₂, Co(NO₃)₂, Cr(NO₃)₃) in 4 mL of H₂O was added with vigorous stirring, giving a deep brown solution. When the dissolution of the transition metal salt was complete, 164 mg (0.6 mmol) of K₂S₂O₈ in 2.5 mL of H₂O was added. The solution was maintained at 90 °C for 60 min. The oxidation is complete after 60 min. KCl (2 g) was added to the hot solution, and the solution was cooled to room temperature. The solution was then placed at 5°C overnight. The resultant purple crystals were collected. Yields are falling between 70% and 90%.

(2) General route to synthesize rare earth metal monosubstituted α_2 -K₇P₂W₁₇O₆₁M (M=RE, Pr/Nd/Gd/Dy/Lu): α_2 -K₁₀P₂W₁₇O₆₁·15H₂O (5 g, 1.017 mmol) was dissolved in 50 mL of 0.5 M sodium acetate buffer at pH=5.5 at 70°C to form a clear solution. 3.08 mmol of LnCl₃ (PrCl₃, Nd(NO₃)₃, GdCl₃, DyCl₃, Lu(NO₃)₃) was dissolved in a minimum amount of water and added dropwise to the stirring K₁₀P₂W₁₇O₆₁ solution. KCl (5 g) was added to the reaction mixture, and the clear solution was cooled in the refrigerator. After 2 h, a white precipitate formed. Yields are falling between 65% and 90%.

The polyhedral representations of two monosubstituted Dawson-type POM isomers are shown in **Supplementary Fig. 2.**

Comment 11. Line 55, a mixture of α_1 and α_2 products, what is the difference between α_1 and α_2 and how are they distinguished?

Reply: Thanks for the comment. α_1 and α_2 are two monovacant Dawson-type polyoxometalate isomers (*J. Am. Chem. Soc.* 1991, **113**, 7209-7221). The difference between α_1 and α_2 is the location of the vacancy. In this study, the monovacant Dawson-type polyoxometalate is α_2 and exhibits a vacancy at the top of polyoxometalate. However, the vacancy of α_1 is near the side position, as shown in **Fig. R3**. In the revised supplementary information, this part has been included in the **Supplementary Fig. 2**.

Fig. R3. Polyhedral representations of two monovacant Dawson-type polyoxometalate isomers α_1 (a) and α_2 (b).

Comment 12. 1.4, in the replacement of K by Ca forming $\text{Ca}_2\text{-P}_2\text{W}_{16}$, what is the role of octadecene and oleylamine in nanowire formation?

Reply: We thank this reviewer for this comment. We want to reply to the reviewer's comments on the following three aspects:

(1) We have synthesized $\text{Ca}_2\text{-P}_2\text{W}_{16}$ SNWs using $\text{H}_6\text{P}_2\text{W}_{18}\text{O}_{62}$ as a raw material. The morphology of SNWs is shown in **Fig. R4**. Consequently, we think that the presence of H^+ or K^+ in this reaction does not have a significant influence on SNW synthesis. Nevertheless, the yield of $\text{Ca}_2\text{-P}_2\text{W}_{16}$ SNWs is higher using $\text{k}_6\text{P}_2\text{W}_{18}\text{O}_{62}$ than $\text{H}_6\text{P}_2\text{W}_{18}\text{O}_{62}$. Hence, in this study, we chose for $\text{k}_6\text{P}_2\text{W}_{18}\text{O}_{62}$.

Fig. R4. Morphologies of $\text{Ca}_2\text{-P}_2\text{W}_{16}$ SNWs synthesized with $\text{H}_6\text{P}_2\text{W}_{18}\text{O}_{62}$ as raw material. **a-b**, STEM images of $\text{Ca}_2\text{-P}_2\text{W}_{16}$ SNWs. **c**, AC-HAADF-STEM image of $\text{Ca}_2\text{-P}_2\text{W}_{16}$ SNW.

(2) Here, we have explained the role of octadecene and oleylamine in SNW formation. **ODE as a solvent, creates an oil-water interface when mixed with water.** Past studies have demonstrated that this interface, where good and bad solvents coexist, can induce the formation of ultrafine structures, including nanowires, nanosheets, nanoribbons, and so on. (*Nat. Chem.* 2022, **14**, 433-440; *Science* 2022, **377**, 100-104; *Nat. Commun.* 2015, **6**, 8756)

(3) **Oleylamine serves two functions in SNW formation.** Firstly, as a surfactant, oleylamine helps the formation of emulsion when ODE is mixed with water. It can stabilize the emulsion and significantly increase the two-phase interface area. Secondly, the protonated oleylamine can be adsorbed to the SNW through electrostatic force as **ligands**, thereby stabilizing the morphology of SNWs. It plays a crucial role in SNW formation.

Comment 13. 1.4, How much is the yield of $\text{Ca}_2\text{-P}_2\text{W}_{16}$?

Reply: Thanks for the comment. We synthesized seven samples by the same experimental method. Yields are falling between **73.3%** and **59.6%**. (**Fig. R5**). We have included this part in the **Supplementary Fig. 3**.

Fig. R5. Photographs of $\text{K}_6\text{P}_2\text{W}_{18}\text{O}_{62} \cdot 14\text{H}_2\text{O}$ powder (a), $\text{Ca}_2\text{-P}_2\text{W}_{16}$ SNWs solid (b-g). Yields are falling between 73.3% and 59.6%.

Comment 14. Line 77, temperature set in the oven needs to be provided.

Reply: Thanks for the comment. We have provided the temperature set in the oven in the revised **Supplementary Information 1.6** as follows:

“When it gelled, a piece of gel was cut from the bulk and put into a centrifugal tube weighing W_1 , and the total weight was recorded as W_2 . Subsequently, put it in an oven and dried the gel at 60°C for 8 h.”

Comment 15. Line 81, Sentence incomplete.

Reply: Thanks for the comment. We have rewritten this part in the revised **Supplementary Information 1.7** as follows:

Line 81: “Gels prepared in the same batch were used for mechanical tests. The mechanical properties of gels were tested through the rheometer (Anton Paar, MCR301).”

Comment 16. Line 153, 162, Reference needed.

Reply: Thanks for the comment. These literatures were cited in the **Supplementary Note 2**. section of the revised supplementary information as follows:

Line 153: “(1) **The interaction energy between SNWs and SNWs.** The interaction energy between SNWs and SNWs can be estimated by the optimal packing model, which assumes that the ligands lying on the surface-surface line pack densely within a narrow volume (**Supplementary Fig. 5**). To the first assumption, the interaction energy can be estimated by two parallel C18 chains. The van der Waals attraction (U_{C18}) between two nearest parallel alkyl chains of length L from N identical basic units ($L=N\lambda$) and separated by a distance D has been given by Salem:⁵ (*J. Chem. Phys.* 1962, **37**, 2100-2113).

$$U_{C18} = A \frac{3\pi}{8\lambda^2} \frac{L}{D^5} \quad (\text{S1})$$

Line 162: “The elastic repulsion energy between two C18 chains can be calculated on the basis of the elastic modulus (E), which is ~1.3 GPa for octadecyl alkyl chains.⁶ (*Appl. Phys. Lett.* 2009, **94**, 131909). Hence the elastic repulsion energy can be estimated to be:

$$U_{el} \approx \frac{1}{2} \times \frac{EA_0}{L} \approx (25.8k_B T) \times (2L - d)^2 \quad (\text{S3})$$

These literatures are included in the revised manuscript as references 32 and 33.

Comment 17. Line 152 and 160, Paralleled should be changed to parallel.

Reply: We thank this reviewer for the suggestion. We have changed “paralleled” to “parallel” in the revised **Supplementary Note 2** as follows:

Line 152: “Assuming that two mutually parallel alkyl chains have maximized van der Waals forces, as indicated by Salem, the attractive interactions between SNWs and solvent can be estimated from the pairs of alkyl chains that are parallel arranged.”

Line 160: “To the first assumption, the interaction energy can be estimated by two parallel C18 chains.”

Response to comments from Reviewer #2

After a careful reading of the manuscript, this research article is recommended for publication in Nature Communications, with some slight remarks and suggestions.

We thank reviewer #2 for the nice comments and hope that our revision (see below) will clarify his/her concerns.

Comment 1. The statistical analysis is not really explained. Please be more precise, show results and discuss them.

Reply: Thanks for the comment. Where reviewers have questions should be in the “Statistical results of SMFS tests (SMFS)” section.

(1) SMFS based on atomic force microscopy (AFM) has been a powerful tool for quantitatively investigating material mechanics at the nanoscale. (Nano Res. 2022, 15, 773-786; Polym. Chem. 2020, 11, 7087-7093; CCS Chem. 2020, 2, 513-523). People often use SMFS to characterize the properties of polymers. They can calculate the strength and flexibility of the molecules from the height and slope of the peaks by analyzing these curves. (Science 1997, 275, 1295-1297) **Firstly, the SMFS test is credible and validated, and secondly, this work focuses on improving the performances of the gels, so we did not introduce statistical analysis in detail in the original version, but we conducted detailed statistics and analysis of the results.** To quantitatively measure the PL of Ca₂-P₂W₁₆ SNWs, we employed contact mode imaging on an Asylum Research Cypher VRS (Figs. 2a-c). The PL calculated by the worm-like chain (WLC) model corresponded to the rigidity of SNWs, with a longer PL, the SNW was more rigid. (Fig. R6., CCS Chem. 2023, 10.31635/ccschem.023.202302729)

Fig. R6. Representative SMFS signals (blue), WLC model fitting curves (red), and schematic core structures (left-upper in each diagram) of SNWs: GdOOH, BiO-PMA, and Ca-PW₁₂. Reprinted with permission from Shi, Y., et al., "Revealing the Flexibility of Inorganic Sub-Nanowires by Single-Molecule Force Spectroscopy", DOI: 10.31635/ccschem.023.202302729, Copyright 2023, Chinese Chemical Society.

(2) In **Fig. 2d**, **2e (Fig. R8)** of the revised manuscript, **each statistical results from 300 samples**. Besides the statistical analysis on the PL and rupture force of Ca₂-P₂W₁₆ SNWs Ca₂-P₂W₁₅M SNWs also be evaluated and counted. Testing and statistical methods have been added to the part of “**1.8 Procedures of Single-molecule force (SMFS) test**” in the revised Supplementary Information. The results of the statistical analysis are shown in **Supplementary Fig. 14 and 15** and **Supplementary Table 7** within the revised Supplementary Information.

In conclusion, through SMFS, we obtained statistical average values for both the PL and rupture force of the SNWs. These values serve two distinct purposes: **1) The PL provides the flexible properties of SNWs; 2) The rupture force determines the interaction between the SNWs and the probe tip (proved to be van der Waals force).**

Comment 2. Please be more precise about the reproducibility and repeatability of each technique used.

Reply: We thank this reviewer for this comment.

(1) For the reproducibility and repeatability tests of mechanical properties, including compression stress-strain testing technique and tensile stress-strain testing technique, we chose gels with a mass fraction of 10.0% as samples. Initially, we synthesized 40 gels using the same batch of SNWs. **Fig. R7a** displays 8 compression stress-strain curves of the samples, while **Fig. R7b** is the histogram of compression stress (kPa) and compression strain (%) at the maximum strain. Gaussian distribution of the maximum compression strain (%) and compression stress (kPa) shown in **Fig. R7c** demonstrates the excellent reproducibility and repeatability of the testing technique. **Figs. R7d-R7f** provides statistical data on tensile stress-strain, further showcasing its reproducibility. This part is also included in **Supplementary Fig. 22** and the revised manuscript.

Fig. R7. The reproducibility and repeatability of mechanical properties of $\text{Ca}_2\text{-P}_2\text{W}_{16}$ SNW-octane gels (10.0%). a-b, Typical compressive stress-strain curves (a) and tensile stress-strain curves (b) of the gels. c-d, Histogram of compression strain (%), compression stress (kPa) (c) and tensile strain (%), tensile stress (d) when eight $1 \times 1 \times 1 \text{ cm}^3$ gels (10.0%) in the same batch had the maximum strain. e-f, Gaussian distribution of the maximum compression strain (%), compression stress (kPa) (e) and tensile strain (%), tensile stress (f) of 40 gel samples.

Fig. R8. Statistical histograms of PL (a) and rupture force (b) of $\text{Ca}_2\text{-P}_2\text{W}_{16}$ SNWs from representative samples of 300, respectively.

(2) In addition, reproducibility data for single-molecule force spectrometry tests of $\text{Ca}_2\text{-P}_2\text{W}_{16}$ SNWs have been provided in **Fig. R8**. This part is included in **Fig. 2d, 2e**. However, the single-molecule force spectrometry testing technique of $\text{Ca}_2\text{-P}_2\text{W}_{15}\text{M}$ ($\text{M}=\text{Fe}, \text{Mn}, \text{Co}, \text{Ni}, \text{Cr}$) SNWs and $\text{Ca}_2\text{-P}_2\text{W}_{15}\text{M}$ ($\text{M}=\text{Pr}, \text{Nd}, \text{Gd}, \text{Dy}, \text{Lu}$) SNWs have been provided in **Fig. R9-R10**. This part is also included in **Supplementary Fig. 14-15**.

Fig. R9. Statistical histograms of PL (left) and rupture force (right) of $\text{Ca}_2\text{-P}_2\text{W}_{15}\text{M}$ (M=Fe, Mn, Ni, Co, Cr) SNWs from representative samples of 300, respectively.

Fig. R10. Statistical histograms of PL (left) and rupture force (right) of $\text{Ca}_2\text{-P}_2\text{W}_{15}\text{M}$ (M=Pr, Nd, Gd, Dy, Lu) SNWs from representative samples of 300, respectively.

Comment 3. Please include the sources of all materials used in the study.

Reply: Thanks for the comment. All materials used in the study have been provided in **Supplementary Methods-1.1 Materials** in the revised supplementary information as follows:

“All chemicals were used as received without any further purification: $\text{Ca}(\text{NO}_3)_2 \cdot 4\text{H}_2\text{O}$ (99.0%), $\text{H}_3\text{PW}_{12}\text{O}_{40} \cdot x\text{H}_2\text{O}$ (PTA, AR), $\text{Na}_2\text{WO}_4 \cdot 2\text{H}_2\text{O}$ (99.5%), HCl (37%), H_3PO_4 (85%), KCl (99.8%), $\text{MnCl}_2 \cdot 4\text{H}_2\text{O}$ (99.9%), $\text{Ni}(\text{NO}_3)_2 \cdot 6\text{H}_2\text{O}$ (99.5%), $\text{Co}(\text{NO}_3)_2 \cdot 6\text{H}_2\text{O}$ (99.5%), $\text{Cr}(\text{NO}_3)_3 \cdot 9\text{H}_2\text{O}$ (99.0%), $\text{PrCl}_3 \cdot 6\text{H}_2\text{O}$ (99.5%), $\text{Nd}(\text{NO}_3)_3 \cdot 6\text{H}_2\text{O}$ (99.5%), $\text{GdCl}_3 \cdot 6\text{H}_2\text{O}$ (99.0%), $\text{DyCl}_3 \cdot 6\text{H}_2\text{O}$ (99.9%), $\text{Lu}(\text{NO}_3)_3 \cdot 6\text{H}_2\text{O}$ (99.5%), ethanol (99.5%), n-hexane (99.0%), n-heptane (99.5%), cyclohexane (99.5%), toluene (99.5%), chloroform (99.0%), absolute ethanol (99.5%), acetone (99.5%), formamide (99.5%), acetonitrile (99.0%) and ethylenediamine (99.0%) were purchased from Sinopharm Chemical Reagent Beijing Co., oleylamine (Sigma-Aldrich, 70%), 1-octadecene (ODE, Sigma-Aldrich, 90%), fluorescein (Sigma-Aldrich, 98%), Nile red (Sigma-Aldrich, 97.5%), perylene diimide (PDI, Aladdin, 95%).”

Comment 4. References should be checked as some references are not properly formatted.

Reply: Thanks for the comment. We have revised all the references in the revised manuscript to conform to the format of *Nature Communications*.

Response to comments from Reviewer #3

In this work, a new class of molecular gelator for organic liquids is reported - i.e. subnanometer inorganic nanowires formed by polyoxometalates clusters, linked by calcium ions. This work explores how these nanowires are formed, at the molecular level, using combinations of electron microscopy and spectroscopy, molecular simulations, and mechanical characterization. It also expands the available library of gelators by substituting W atoms by Fe or Pr, for example. Three main results stand out: (1) the capacity to probe the strength of fibers at the nm level using AFM; (2) the very high quality microscopy analysis coupled with the molecular simulations, offering a clear perspective on how these fibers are formed; (3) the significant mechanical properties that the gel displays - most notably their elasticity.

We thank reviewer #3 for the nice comments and hope that our revision (see below) will clarify his/her concerns.

Comment 1. In my opinion, the experimental work is of very high quality, and the molecular simulations truly bring additional elements of comprehension. However, I do have some reserve at this point regarding the novelty of the results, their importance for a large scientific audience, and their impact. The gelator itself that is the object of this work - mostly $\text{Ca}_2\text{-P}_2\text{W}_{16}$ - appears to have been reported previously by the authors in Ref.9. In this work, a number of features, including the fiber-like structure and mechanical properties, are presented. It is not clear to me then how the results in the current submitted work differ or expand on those presented in Ref.9 or bring truly new insight.

Reply: We sincerely thank this reviewer for pointing these issues out. We apologize for not clearly showing the significance and innovativeness of this work in advancing the field of organogels. **The work of Ref. 9. is the first and only one to use SNWs to prepare elastic and free-standing organogels, which is a pioneering work. Although this SNW-based organogel can solve many problems, the component and intrinsic properties of SNWs need to be further optimized. What's more, the mechanical properties and functionality of SNW-based organogels need to be further expanded. The work in this paper is based on the above work (Ref. 9).** The significance and

expansion of this work, we will provide the following three explanations:

(1) **We changed the ligand density, flexibility, and CGC of SNWs by adjusting the type of POM cluster and the linked mode of units. In addition, the $\text{Ca}_2\text{-P}_2\text{W}_{16}$ SNW has never been reported previously in Ref.9.** and distinguishes itself from the Ca-PW_{12} SNW (*Science* 2022, **377**, 100-104) in several significant aspects: **1)** Firstly, their construction varies markedly. The $\text{Ca}_2\text{-P}_2\text{W}_{16}$ SNW is constructed from two Ca^{2+} -bridged Dawson-type polyoxometalate clusters, while Ca-PW_{12} SNWs consist of one Ca^{2+} ion bridging two Keggin-type POM clusters. Because of the higher charge of the $\text{Ca}_2\text{-P}_2\text{W}_{16}$ SNWs, ligand density on their surface is three times that of the Ca-PW_{12} SNWs; **2)** The structural differences, such as variances in polyoxometalate clusters and cluster connectivity methods, lead to significant variations in SNW flexibility. We employed single-molecule force spectroscopy (SMFS) results to elucidate molecular-level differences between these SNWs. We think the change from one Ca^{2+} ion to two at the joint restricts the flexibility of SNWs. Moreover, the Dawson-type POM cluster has a larger size compared to the Keggin-type POM cluster (as depicted in **Fig. R.11**). This constitutes a significant factor contributing to the increase in the PL of $\text{Ca}_2\text{-P}_2\text{W}_{16}$ SNWs; **3)** The higher surface ligand density in $\text{Ca}_2\text{-P}_2\text{W}_{16}$ SNWs, compared to Ca-PW_{12} SNWs, enhances the interaction between the SNWs and organic solvents. Consequently, $\text{Ca}_2\text{-P}_2\text{W}_{16}$ SNWs exhibit an improved solvent-locking ability, leading to a lower CGC of 0.28%, as opposed to 0.53% for Ca-PW_{12} SNWs in octane. **In summary, numerous distinctions exist between the $\text{Ca}_2\text{-P}_2\text{W}_{16}$ SNWs in this study and the Ca-PW_{12} SNWs in Ref. 9. It encompassed variations in constituent, flexibility, ligand density, and CGC of SNW. These distinctions highlight the divergence between the two works.**

Fig. R11. Size comparison of different building blocks of Ca-PW_{12} SNWs and $\text{Ca}_2\text{-P}_2\text{W}_{16}$ SNWs.

(2) **Based on Ref. 9, this work regulates the mechanical properties of SNWs-based organogels through the regulation of microstructure (POM units and linked mode).** Microstructure changes in SNWs lead to enhanced mechanical properties in $\text{Ca}_2\text{-P}_2\text{W}_{16}$ SNWs-based organogels. These differences in mechanical properties between the two gels were presented in **Table R3** and **Fig. R12**, which are included in the revised **Supplementary Table 5-6**. In this study, we significantly enhanced the mechanical properties of organic gels, including compressive strength, tensile strength, and yield point. This work reveals the relationships between the microstructures of inorganic gelators and the properties of organogels. This part is included in the revised manuscript.

Table R3. Mechanical properties of $\text{Ca}_2\text{-P}_2\text{W}_{16}$ SNWs-based gels and Ca-PW_{12} SNWs-based gels

Samples	Mass fraction (%)	Tensile strength (kPa)	Compressive strength (kPa)	Yield point (MPa)
$\text{Ca}_2\text{-P}_2\text{W}_{16}$	10.0%	29.0	34.5	0.0273
$\text{Ca}_2\text{-P}_2\text{W}_{16}$	8.3%	23.1	25.5	0.0196
$\text{Ca}_2\text{-P}_2\text{W}_{16}$	6.2%	16.6	12.5	0.0122
$\text{Ca}_2\text{-P}_2\text{W}_{16}$	3.0%	10.5	4.9	0.0071
Ca-PW_{12}	10.0%	24.2	26.1	0.0209
Ca-PW_{12}	8.0%	19.7	20.2	0.0163
Ca-PW_{12}	6.3%	16.5	17.8	0.0116
Ca-PW_{12}	3.0%	7.9	7.0	0.0061

Fig. R12. Histogram of the tensile strength, compressive strength, and yield point of Ca-PW₁₂ SNWs-based gel and Ca₂-P₂W₁₆ SNWs-based gel.

(3) **The expansion of organogel functions.** Functionalized organogel hybrid materials offer promising applications in various fields due to their distinctive properties. One example is that organogels were used to design light-harvesting assemblies. (*Chem. Soc. Rev.* 2008, **37**, 109-122.) The organic phase in polymer-based organogel exhibits dynamic characteristics rather than being static. (*Langmuir* 2015, **31**, 13850-13859; *J. Control. Release.* 2018, **271**, 1-20.) **Currently, SNW-based organogels meet some of the requirements for practical applications. However, enriching the properties and functions of SNW-based organogels is an opportunity with great challenges.** Due to the limited literature on SNW-based organogels, research programs to develop more functional organogels face great challenges. So, we need to do more experiments and accumulate experience. We propose that Ca₂-P₂W₁₆ SNW-based gel can also exhibit similar dynamic properties to liquid solvents. In addition, incorporating fluorescent dyes or fluorophores into an organogel and stimulating these dyes or groups to lead photoluminescence of Ca₂-P₂W₁₆ SNWs-based gels. 1) Firstly, we opted for neutral fluorescent molecules. This allowed the FOMs to partially substitute for octane and

engage with oleylamine through van der Waals force; 2) Secondly, fluorescent molecules have the advantage of displaying color, enabling us to observe the diffusion of molecules within the gel without compromising the integrity of the SNWs or the gel structure; 3) Thirdly, owing to the relatively weak van der Waals force, the FOMs can be eventually separated from the gel, allowing for the recycling of both the FOMs and the SNWs. Indeed, the utilization of other organogel as a selective adsorbent for removing dye molecules has been achieved. (*Langmuir* 2021, **37**, 3996-4006; *Mater. Lett.* 2020, **272**, 127854.) In addition, FOMs were used as a kind of functional molecule to prepare fluorescent organogels. Therefore, SNW-based organogels may be endowed with more functions through the introduction of other functional organic molecules, such as photosensitive, conductive, catalytic, and drug organic molecules, etc., or inorganic nanocrystals. This part has been included in **Fig. 5** of the revised manuscript.

This work guides the preparation of high-performance, functional organogels, and explores the connections between the microstructures of inorganic gelators and the properties of organogels.

Comment 2. The results in Figure 2, about the Single-molecule force spectroscopy, also seem to expand on those presented in Ref.10 by the same authors. It is not clear how the results presented here really bring new original insight, or if it is a refinement.

Reply: Thanks for the comment. Single-molecule force spectroscopy (SMFS) is a powerful experimental technique used to investigate the mechanical properties of individual molecules and their interactions at the nanoscale. In the field of nanotechnology, SMFS is used to characterize and manipulate individual nanoscale structures, such as nanoparticles, nanowires, and nanotubes. It aids in assessing their mechanical properties and potential applications in materials science and electronics. Advantages of SMFS include the ability to study individual molecules, providing insights not attainable with bulk measurements; High sensitivity, allowing the detection of weak molecular interactions; Precise control over the applied forces, enabling the investigation of a wide range of mechanical properties. **In Ref. 10., SMFS has been used to characterize and calculate the flexibility of inorganic SNWs for the first time. As a reliable testing technique, it is also used to characterize the flexibility of the new SNW in this work.**

Fig. R13. Histogram of the PLs of SNWs, namely Ca-PW₁₂, Ca₂-P₂W₁₆, Ca₂-P₂W₁₅M (M=TM, RE) SNWs.

Through SMFS, we obtained statistical average values for both the **PL** and **rupture force** of the SNWs. These values serve two distinct purposes: 1) **The PL provides the flexible properties of SNWs**; 2) **The rupture force determines the interaction between the SNWs and the probe tip (proved to be van der Waals force)**. Compared with Ca-PW₁₂ SNWs, the flexibility of Ca₂-P₂W₁₆ and Ca₂-P₂W₁₅M (M=TM, RE) SNWs, revealed generally low. This can be attributed to the presence of two Ca²⁺ at the joint reduced the flexibility of the SNW. The comparison of the PLs of the Ca-PW₁₂ SNWs and Ca₂-P₂W₁₅M (M=TM, RE) SNWs is illustrated in **Fig. R13** and **Table R4**. In summary, we changed the flexibility of SNWs by adjusting the type of polyoxometalates cluster, leading to improved mechanical properties of SNW-based organogels. This study effectively established a structure-function relationship between microstructure and macroscopic properties. Consequently, the SMFS test is valuable for enhancing our understanding of the intrinsic properties of various SNWs. This part is also included in the revised **Supplementary Fig. 18** and **Supplementary Table 7**. This part is also included in the revised manuscript.

Table R4. The persistence lengths and rupture forces of SNWs and other materials

	Samples	Persistence length (nm)	Rupture force (pN)
DNA	ss DNA ^a	4	--
	ds DNA ^b	50	--
Covalent bond	Silicon-carbon bond ^c	--	2000±300
	Sulfur-gold anchor ^c	--	1400±300
SNWs	GdOOH ^d	<10	50-200
	BiO-PMA ^d	<10	50-200
	Ca-PW ₁₂ ^d	0.22±0.08 ^a	50-200
	Ca ₂ -P ₂ W ₁₆	1.13±0.31	131.6±21.7
	Ca ₂ -P ₂ W ₁₅ Fe	0.58±0.28	162.5±25.0
	Ca ₂ -P ₂ W ₁₅ Mn	0.53±0.22	150.6±24.3
	Ca ₂ -P ₂ W ₁₅ Co	0.71±0.29	172.2±27.9
	Ca ₂ -P ₂ W ₁₅ Ni	0.79±0.29	143.6±21.9
	Ca ₂ -P ₂ W ₁₅ Cr	0.69±0.29	139.8±19.8
	Ca ₂ -P ₂ W ₁₅ Pr	0.93±0.18	294.3±25.6
	Ca ₂ -P ₂ W ₁₅ Nd	1.08±0.18	253.3±20.8
	Ca ₂ -P ₂ W ₁₅ Gd	1.13±0.20	276.3±24.2
	Ca ₂ -P ₂ W ₁₅ Dy	1.21±0.23	256.5±28.9
Ca ₂ -P ₂ W ₁₅ Lu	1.12±0.22	284.8±26.3	

^a *Macromolecules* 1997, **30**, 5763-5765; ^b *Science* 1994, **265**, 1599-1600; ^c *Science* 1999, **283**, 1727-1730; ^d *CCS Chem.* 2023, 10.31635/ccschem.023.202302729.

Comment 3. My understanding at this point then is that the new content is really about tuning the properties of the gelator and resulting gel by substituting W atoms with transition

metals (herein Fe, for example) or rare earths (RE). It seems however that the main figures are mostly concerned with the basic gelator, whereas the results with the substituted gelators are presented as Supporting information.

Reply: We thank this reviewer for the suggestion. However, we apologize for the lack of clarity in the original manuscript, which has resulted in some difficulties for the reviewer in understanding it. We fabricated the $\text{Ca}_2\text{-P}_2\text{W}_{16}$ and $\text{Ca}_2\text{-P}_2\text{W}_{15}\text{M}$ (M =Fe, Mn, Ni, Co, Cr, Pr, Nd, Gd, Dy, Lu) SNWs, which were then employed for preparing organogels. The mechanical performance and stability of prepared organogels are improved due to the enhanced interactions between SNWs and locked organic molecules. **In this work, the primary innovation lies in $\text{Ca}_2\text{-P}_2\text{W}_{16}$ SNWs and the corresponding $\text{Ca}_2\text{-P}_2\text{W}_{16}$ SNW-based organogels, while $\text{Ca}_2\text{-P}_2\text{W}_{15}\text{M}$ (M =Fe, Mn, Ni, Co, Cr, Pr, Nd, Gd, Dy, Lu) SNWs and the $\text{Ca}_2\text{-P}_2\text{W}_{15}\text{M}$ SNW-based organogels serve a supporting role.** We explained it from two aspects:

(1) Firstly, to assess the universality of the synthesis method and the properties of the $\text{Ca}_2\text{-P}_2\text{W}_{16}$ and $\text{Ca}_2\text{-P}_2\text{W}_{15}\text{M}$ (M=TM, RE) SNWs, we modify the properties of the gelator and the resulting gel by replacing tungsten atoms with transition-metals (TM) or rare-earth elements (RE). Make this work more convincing.

(2) Secondly, the incorporation of transition metal (TM) can confer photocatalytic or electrocatalytic properties upon $\text{Ca}_2\text{-P}_2\text{W}_{16}$ SNWs, as exemplified by the CO_2 reduction catalyzed by high-entropy sub-nano materials in our group. (*J. Am. Chem. Soc.* 2021, **143**, 16217-16225; *Small Struct.* 2022, 2200039; *Nat. Chem.* 2019, **11**, 839-845.) This example is not unique. Furthermore, the introduction of the rare-earth element (RE) can also enhance the performance of SNW-based organic catalysis. (*Nat. Chem.* 2022, **14**, 433-440.) Additionally, the rare-earth elements have distinctive magnetic, optical, and electrochemical properties that can endow SNWs with exceptional magnetic, optical, and electrical properties. These attributes have the potential to significantly enhance SNW performance and broaden their range of applications.

To enhance the clarity and comprehensibility of the article, we have significantly revised the abstract and introduction of the revised manuscript. The revised

manuscript ensures that the primary information and supporting information are more accessible to readers.

Comment 4. I am not convinced at this point then that the content fits the scope of Nature Communications - it would be better suited I believe for a high quality more specialized journal. In addition, the beginning of the Introduction is quite generic, and it is not clear at all why this work is important for the broad readership of Nature Communications-needs to be better explained and put in context. However, I would be open to consider the arguments of the authors, and to reconsider my recommendation if need be.

Reply: We thank reviewer #3 for the comments. This is very important for us. We agree that the beginning of the Introduction is generic. We replied to the reviewer's comments as follows:

Ca-PW₁₂ SNW-based organogels are facile, secure, and energy-efficient. (Science 2022, 377, 100-104) However, **the limited charge of the PW₁₂ cluster unit restricts the mechanical performance and functions of SNW-based organogels constructed using it.** In the SNWs-based organogels, the interaction between the solvent molecules and surface ligands directly affects the performance of organogels. (Science 2022, 377, 100-104.; ACS Nano 2023, 17, 20-26.) Using the highly charged POM clusters as building blocks for constructing the SNWs can effectively regulate their surface ligand density, and then improve the mechanical performance of SNW-based organogels and get low CGC. TGA results showed a threefold rise in ligand density on the SNW surface (**Fig. R14, Supplementary Fig. 4, and Supplementary Note 1**), resulting in a substantial enhancement in the macroscopic mechanical properties of the SNW-based organogels (**Table R3 and Fig. R12**). In addition, these SNWs effectively enhance the interaction between ligand molecules and solvent molecules, resulting in an obvious reduction in CGC (0.28%).

Fig. R14 Thermogravimetric analysis (TGA) on Ca-PW₁₂ SNWs and Ca₂-P₂W₁₆ SNWs.

In this work, we focus on highlighting the hierarchical relationship among “highly charged clusters”, “high ligand density SNWs” and the resulting “improved mechanical properties of SNW-based organogels”. The progressive relationship between these three parts is not clearly and comprehensively introduced in the Introduction of the original manuscript. **Considering the cohesion of context, we revised the Introduction by dividing it into three paragraphs. The introductions of "subnanometer inorganic nanowires (SNWs)" and "polyoxometalate (POM) clusters" have been added, and the relationship between the Introduction and the context has been reorganized in the Introduction of the revised manuscript.**

We appreciate the reviewer for the nice comments again and hope our responses will address your concerns.

REVIEWERS' COMMENTS

Reviewer #1 (Remarks to the Author):

Reviewer Comment for Manuscript ID: NCOMMS-23-31741A

Manuscript Title: "Sub-nanowires formed by highly charged polyoxometalate clusters for improving mechanical properties of organogels"

I have thoroughly reviewed the authors' responses and find that the revisions have notably enhanced the quality and clarity of the manuscript. The authors have addressed all the comments given by me. The efforts put into refining the content, structure, and language are commendable. I am confident that this article will offer a substantial and valuable contribution to the research community in organogels. I am sure that this work aligns with the high standards and goals of Nature Communications, and I am inclined to recommend its acceptance for publication in the journal.

Reviewer #2 (Remarks to the Author):

After a careful proof-reading of the manuscript, this research article is recommended for publication in Nature Communications, with some slight remarks and suggestions.

- The statistical analysis is not really explained. Please be more precise, show results and discuss them.
- Please be more precise about the reproducibility and repeatability of each technique used.
- Please include the sources of all materials used in the study.
-
- References should be checked as some references are not properly formatted

Reviewer #3 (Remarks to the Author):

I have reviewed the answers and modifications to the manuscript provided by the authors. My main concern previously was about the originality of the work compared to other previously published results. The authors have carefully explained, in great details, how the present work distinguishes itself from their previous publications on the subject of SNW's: (1) the differences in chemistry between previous Ca-PW12 based SNWs, and new ones based on Ca₂P₂W₁₆; (2) the enhanced properties obtained by using Ca₂P₂W₁₆; (3) the capacity to tune and expand the library of SNW by analyzing the effect of other metals (M). They have also modified the manuscript to emphasize these original contributions compared to their previous work. Again, I also underline the quality of their experimental work. I do support then publication in Nature Communications, after proofreading the manuscript to correct the remaining typos.

Reviewer #1 (Remarks to the Author):

I have thoroughly reviewed the authors' responses and find that the revisions have notably enhanced the quality and clarity of the manuscript. The authors have addressed all the comments given by me. The efforts put into refining the content, structure, and language are commendable. I am confident that this article will offer a substantial and valuable contribution to the research community in organogels. I am sure that this work aligns with the high standards and goals of Nature Communications, and I am inclined to recommend its acceptance for publication in the journal.

Reply: We thank the reviewers for their valuable comments, and we are glad that the reviewers were satisfied with our revisions.

Reviewer #2 (Remarks to the Author):

After a careful proof-reading of the manuscript, this research article is recommended for publication in Nature Communications, with some slight remarks and suggestions.

Reply: We thank the reviewers for their valuable comments.

1. The statistical analysis is not really explained. Please be more precise, show results and discuss them.

Reply: Thanks for the comment. The persistence lengths of $\text{Ca}_2\text{-P}_2\text{W}_{15}\text{TM}$ NWs and $\text{Ca}_2\text{-P}_2\text{W}_{15}\text{RE}$ NWs are 0.53-0.79 nm and 0.93-1.21 nm, respectively (**Fig. S1-S4**). Thus, the flexibilities of $\text{Ca}_2\text{-P}_2\text{W}_{16}$ and $\text{Ca}_2\text{-P}_2\text{W}_{15}\text{M}$ NWs are generally lower than that of Ca-PW_{12} NWs. In addition, we found the rupture forces of $\text{Ca}_2\text{-P}_2\text{W}_{15}\text{RE}$ NWs (253.3-294.3 pN) are greater than that of $\text{Ca}_2\text{-P}_2\text{W}_{16}$ NWs and $\text{Ca}_2\text{-P}_2\text{W}_{15}\text{TM}$ NWs. These statistical results proved our conjecture that the flexibility is related to the building blocks or connection mode of the NWs. We provided a more detailed explanation of the statistical analysis. And this part is added in the revised Supplementary Information.

Fig. S1 Statistical histograms of persistence lengths and rupture forces of $\text{Ca}_2\text{-P}_2\text{W}_{15}\text{M}$ ($\text{M}=\text{Fe}, \text{Mn}, \text{Ni}, \text{Co}, \text{Cr}$) NWs. These statistical histograms were obtained from representative samples of 300, respectively.

Fig. S2 Statistical histograms of persistence lengths and rupture forces of $\text{Ca}_2\text{-P}_2\text{W}_{15}\text{M}$ (M=Pr, Nd, Gd, Dy, Lu) NWs. These statistical histograms were obtained from representative samples of 300, respectively.

Fig. S3 Histogram of the persistence lengths of NWs, namely Ca-PW₁₂, Ca₂-P₂W₁₆, and Ca₂-P₂W₁₅M (M=TM, RE) NWs.

Fig. S4 Histogram of rupture forces of Ca-PW₁₂, Ca₂-P₂W₁₆, and Ca₂-P₂W₁₅M (M=TM, RE) NWs.

2. Please be more precise about the reproducibility and repeatability of each technique used.

Reply: Thanks for the comment. The statistical results of mechanical properties, including compression stress-strain testing technique and tensile stress-strain testing technique, showed that the reproducibility and repeatability of mechanical properties testing technique for $\text{Ca}_2\text{-P}_2\text{W}_{16}$ NW-octane gels, with a standard deviation range of less than 10%. This part is also included in the revised Supplementary Information.

Fig. S4 The reproducibility and repeatability of mechanical properties of $\text{Ca}_2\text{-P}_2\text{W}_{16}$ NW-octane gels. **a, c** Typical compressive stress-strain curves (**a**) and tensile stress-strain curves (**c**) of the gels (10.0%, samples1-8). **b, d** Histogram of compression strain (% , black bar), compression stress (kPa, red bar) (**b**) and tensile strain (% , black bar), tensile stress (kPa, red bar) (**d**) when eight $1 \times 1 \times 1 \text{ cm}^3$ gels (10.0%, samples1-8) in the same batch had the maximum strain. **e** The statistical results of repeatability of tensile strength and compressive strength dates of 40 $\text{Ca}_2\text{-P}_2\text{W}_{16}$ NW-octane gel samples.

3. Please include the sources of all materials used in the study.

Reply: Thanks for the comment. All materials used in the study have been provided in **Supplementary Methods-1.1 Materials** in the revised supplementary information as follows:

“All chemicals were used as received without any further purification: $\text{Ca}(\text{NO}_3)_2 \cdot 4\text{H}_2\text{O}$ (99.0%), $\text{H}_3\text{PW}_{12}\text{O}_{40} \cdot x\text{H}_2\text{O}$ (PTA, AR), $\text{Na}_2\text{WO}_4 \cdot 2\text{H}_2\text{O}$ (99.5%), HCl (37%), H_3PO_4 (85%), KCl (99.8%), $\text{MnCl}_2 \cdot 4\text{H}_2\text{O}$ (99.9%), $\text{Ni}(\text{NO}_3)_2 \cdot 6\text{H}_2\text{O}$ (99.5%), $\text{Co}(\text{NO}_3)_2 \cdot 6\text{H}_2\text{O}$ (99.5%), $\text{Cr}(\text{NO}_3)_3 \cdot 9\text{H}_2\text{O}$ (99.0%), $\text{PrCl}_3 \cdot 6\text{H}_2\text{O}$ (99.5%), $\text{Nd}(\text{NO}_3)_3 \cdot 6\text{H}_2\text{O}$ (99.5%), $\text{GdCl}_3 \cdot 6\text{H}_2\text{O}$ (99.0%), $\text{DyCl}_3 \cdot 6\text{H}_2\text{O}$ (99.9%), $\text{Lu}(\text{NO}_3)_3 \cdot 6\text{H}_2\text{O}$ (99.5%), ethanol (99.5%), hexane (99.0%), heptane (99.5%), cyclohexane (99.5%), toluene (99.5%), chloroform (99.0%), absolute ethanol (99.5%), acetone (99.5%), formamide (99.5%), acetonitrile (99.0%) and ethylenediamine (99.0%) were purchased from Sinopharm Chemical Reagent Beijing Co., oleylamine (Sigma-Aldrich, 70%), octadecene (ODE, Sigma-Aldrich, 90%), fluorescein (Sigma-Aldrich, 98%), Nile red (Sigma-Aldrich, 97.5%), perylene diimide (Aladdin, 95%). V-shaped Si_3N_4 AFM cantilevers (Bruker, MLCT, Beijing), silicon wafer (GinSRC, Beijing), carbon film coated grids (200 mesh) (EMCN, Beijing).”

4. References should be checked as some references are not properly formatted

Reply: Thanks for the comment. We have revised all the references in the revised manuscript to conform to the format of *Nature Communications*.

Reviewer #3 (Remarks to the Author):

I have reviewed the answers and modifications to the manuscript provided by the authors. My main concern previously was about the originality of the work compared to other previously published results. The authors have carefully explained, in great details, how the present work distinguishes itself from their previous publications on the subject of SNW's: (1) the differences in chemistry between previous Ca- PW_{12} based SNWs, and new ones based on $\text{Ca}_2\text{P}_2\text{W}_{16}$; (2) the enhanced properties obtained by using $\text{Ca}_2\text{P}_2\text{W}_{16}$; (3) the capacity to tune and expand the library of SNW by analyzing the effect of other metals (M). They have also modified the manuscript to emphasize these original contributions compared to their previous work. Again, I also underline the quality of their

experimental work. I do support then publication in Nature Communications, after proofreading the manuscript to correct the remaining typos.

Reply: We thank the reviewers for their valuable comments, and we are glad that the reviewers were satisfied with our revisions.